# Proofreading through spatial gradients

**Vahe Galstyan[1], Kabir Husain[2], Fangzhou Xiao[3], Arvind Murugan[2]\*, Rob Phillips[3,4]\***

[1]Biochemistry and Molecular Biophysics Option, California Institute of Technology, Pasadena, United States; [2]Department of Physics and the James Franck Institute, University of Chicago, Chicago, United States; [3]Division of Biology and Biological Engineering, California Institute of Technology, Pasadena, United States; [4]Department of Physics, California Institute of Technology, Pasadena, United States

**Abstract** Key enzymatic processes use the nonequilibrium error correction mechanism called kinetic proofreading to enhance their specificity. The applicability of traditional proofreading schemes, however, is limited because they typically require dedicated structural features in the enzyme, such as a nucleotide hydrolysis site or multiple intermediate conformations. Here, we explore an alternative conceptual mechanism that achieves error correction by having substrate binding and subsequent product formation occur at distinct physical locations. The time taken by the enzyme–substrate complex to diffuse from one location to another is leveraged to discard wrong substrates. This mechanism does not have the typical structural requirements, making it easier to overlook in experiments. We discuss how the length scales of molecular gradients dictate proofreading performance, and quantify the limitations imposed by realistic diffusion and reaction rates. Our work broadens the applicability of kinetic proofreading and sets the stage for studying spatial gradients as a possible route to specificity.

**\*For correspondence:**
amurugan@uchicago.edu (AM);
phillips@pboc.caltech.edu (RP)

**Competing interests:** The authors declare that no competing interests exist.

## Introduction

The nonequilibrium mechanism called kinetic proofreading (*Hopfield, 1974*; *Ninio, 1975*) is used for reducing the error rates of many biochemical processes important for cell function (e.g. DNA replication [*Kunkel, 2004*], transcription [*Sydow and Cramer, 2009*], translation [*Rodnina and Wintermeyer, 2001*; *Ieong et al., 2016*], signal transduction [*Swain and Siggia, 2002*], or pathogen recognition [*McKeithan, 1995*; *Goldstein et al., 2004*; *Cui and Mehta, 2018*]). Proofreading mechanisms operate by inducing a delay between substrate binding and product formation via intermediate states for the enzyme–substrate complex. Such a delay gives the enzyme multiple chances to release the wrong substrate after initial binding, allowing far lower error rates than what one would expect solely from the binding energy difference between right and wrong substrates.

Traditional proofreading schemes require dedicated molecular features such as an exonuclease pocket in DNA polymerases (*Kunkel, 2004*) or multiple phosphorylation sites on T-cell receptors (*McKeithan, 1995*; *Goldstein et al., 2004*); such features create intermediate states that delay product formation (*Figure 1a*) and thus allow proofreading. Additionally, since proofreading is an active nonequilibrium process often involving near–irreversible reactions, the enzyme typically needs to have an ATP or GTP hydrolysis site to enable the use of energy supplies of the cell (*Yamane and Hopfield, 1977*; *Rodnina and Wintermeyer, 2001*). Due to such stringent structural requirements, the number of confirmed proofreading enzymes is relatively small. Furthermore, generic enzymes without such dedicated features are assumed to not have active error correction available to them.

In this work, we propose an alternative scheme where the delay between initial substrate binding and product formation steps is achieved by separating these events in space. If substrates are spatially localized and product formation is favorable only in a region of low substrate concentration where an activating effector is present then the time taken by the enzyme–substrate complex to

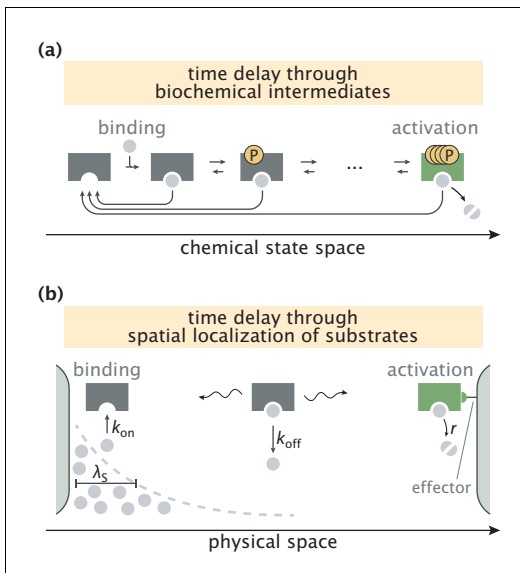

**Figure 1.** Error correction schemes that operate by delaying product formation. (**a**) The traditional proofreading scheme with multiple biochemically distinct intermediates, transitions between which are typically accompanied by energy–consuming reactions. The T-cell activation mechanism with successive phosphorylation events is used for demonstration (**McKeithan, 1995**; **Cui and Mehta, 2018**). (**b**) The spatial proofreading scheme where the delay between binding and catalysis is created by constraining these events to distinct physical locations. The wavy arrows stand for the diffusive motion of the complex. Binding events primarily take place on the length scale $\lambda_s$ of substrate localization.

travel from one location to the other can be used to discard the wrong substrates, which are assumed to unbind from the enzyme more readily than the right substrates (**Figure 1b**). When this delay is longer than substrate unbinding time scales, very low error rates of product formation can be achieved, allowing this spatial proofreading scheme to outperform biochemical mechanisms with a finite number of proofreading steps.

In contrast to traditional proofreading, the nonequilibrium mechanism here does not require any direct energy consumption by the enzyme or substrate itself (e.g. through ATP hydrolysis). This liberates the enzyme from any proofreading-specific molecular features; indeed, any 'equilibrium' enzyme with a localized effector can proofread using our scheme if appropriate concentration gradients of the substrates or enzymes are set up. In this way, the energetic and structural requirements of proofreading can be outsourced from the enzyme and substrate to the gradient maintaining mechanism. It also means that spatial proofreading is easy to overlook in experiments, and that the fidelity of reconstituted reactions in vitro could be lower than the fidelity in vivo.

The lack of reliance on structure makes spatial proofreading more adaptable. We study how tuning the length scale of concentration gradients can trade off error rate against speed and energy consumption on the fly. In contrast, traditional proofreading schemes rely on nucleotide chemical potentials, for example, the out of equilibrium [ATP]/[ADP] ratio in the cell, and cannot modulate their operation without broader physiological disruptions.

Our proposed scheme can be leveraged for specificity if appropriate concentration gradients are set. Such gradients arise in multiple cellular contexts (e.g. near the nucleus, the plasma membrane, the Golgi apparatus, the endoplasmic reticulum [ER], kinetochores, microtubules [**Bivona et al., 2003**; **Caudron et al., 2005**; **Kholodenko, 2006**]) and several gradient-forming mechanisms have been discussed in the literature (**Wu et al., 2018**; **Kholodenko, 2006**; **Kholodenko, 2003**). We conclude our analysis of spatial proofreading by quantifying its limitations as set by realistic reaction rates and gradient formation mechanisms, and discuss examples from the literature, including the localization of mRNAs in polarised cells, and the non-vesicular transport of lipids in eukaryotic cells, in which this mechanism might be in play. Our work motivates a detailed investigation of spatial structures and compartmentalization in living cells as possible delay mechanisms for proofreading enzymatic reactions.

## Results

### Slow transport of enzymatic complex enables proofreading

Our proposed scheme is based on spatially separating substrate binding and product formation events for the enzyme (**Figure 1b**). Such a setting arises naturally if substrates are spatially localized by having concentration gradients in a cellular compartment. Similarly, an effector needed for product formation (e.g. through allosteric activation) may have a spatial concentration gradient localized

elsewhere in that compartment. To keep our model simple, we assume that the right (R) and wrong (W) substrates have identical concentration gradients of length scale $\lambda_S$ but that the effector is entirely localized to one end of the compartment, for example via membrane tethering. In Appendix 4, we extend our study of model performance to the scenario where the two substrates have different localization length scales.

We model our system using coupled reaction–diffusion equations for the substrate-bound ('ES' with $S = R, W$) and free ('E') enzyme densities, namely,

$$\frac{\partial \rho_{ER}}{\partial t} = D \frac{\partial^2 \rho_{ER}}{\partial x^2} - k_{off}^R \rho_{ER} + k_{on} \rho_R \rho_E, \tag{1}$$

$$\frac{\partial \rho_{EW}}{\partial t} = D \frac{\partial^2 \rho_{EW}}{\partial x^2} - k_{off}^W \rho_{EW} + k_{on} \rho_W \rho_E, \tag{2}$$

$$\frac{\partial \rho_E}{\partial t} = D \frac{\partial^2 \rho_E}{\partial x^2} + \sum_{S=R,W} k_{off}^S \rho_{ES} - \sum_{S=R,W} k_{on} \rho_S \rho_E. \tag{3}$$

Here, $D$ is the enzyme diffusion constant, $k_{on}$ and $k_{off}^S$ (with $k_{off}^W > k_{off}^R$) are the substrate binding and unbinding rates, respectively, and $\rho_S(x) \sim e^{-x/\lambda_S}$ is the spatially localized substrate concentration profile which we take to be exponentially decaying, which is often the case for profiles created by cellular gradient formation mechanisms (*Driever and Nüsslein-Volhard, 1988*; *Brown and Kholodenko, 1999*). We limit our discussion to this one-dimensional setting of the system, though our treatment can be generalized to two and three dimensions in a straightforward way.

The above model does not explicitly account for several effects relevant to living cells, such as depletion of substrates or distinct diffusion rates for the free and substrate-bound enzymes. More importantly, it does not account for the mechanism of substrate gradient formation. We analyze a biochemically detailed model with this latter feature and experimentally constrained parameters later in the paper. Here, we proceed with the minimal model above for explanatory purposes. To identify the key determinants of the model's performance, we assume throughout our analysis that the amount of substrates is sufficiently low that the enzymes are mostly free with a roughly uniform profile (i.e. $\rho_E \approx \text{constant}$). This assumption makes *Equations (1-3)* linear and allows us to solve them analytically at steady state. We demonstrate in Appendix 5 that proofreading is, in fact, most effective under this assumption and discuss the consequences of having high substrate amounts on the performance of the scheme.

In our simplified picture, enzyme activation and catalysis take place upon reaching the right boundary at a rate $r$ that is identical for both substrates. Therefore, the density of substrate–bound enzymes at the right boundary can be taken as a proxy for the rate of product formation $v_S$, since

$$v_S = r \rho_{ES}(L), \tag{4}$$

where $L$ is the size of the compartment. In order to keep the analytical results concise and intuitive, we perform our main analyses under the assumption that catalysis is slow, mirroring the study of traditional proofreading schemes (*Hopfield, 1974*). In Appendix 3, we derive the precise conditions under which this treatment is valid, and generalize our analysis to arbitrary catalysis rates.

To demonstrate the proofreading capacity of the model, we first analyze the limiting case where substrates are localized to the left end of the compartment ($\lambda_S \to 0$). In this limit, the fidelity $\eta$, defined as the number of right products formed per single wrong product, becomes

$$\eta = \frac{v_R}{v_W} = \sqrt{\eta_{eq}} \frac{\sinh\left(\sqrt{\tau_D k_{off}^W}\right)}{\sinh\left(\sqrt{\tau_D k_{off}^R}\right)}, \tag{5}$$

where $\eta_{eq} = k_{off}^W / k_{off}^R$ is the equilibrium fidelity, and $\tau_D = L^2/D$ is the characteristic time scale of diffusion across the compartment (see Appendix 1 for the derivation).

*Equation 5* is plotted in *Figure 2* for a family of different parameter values. As can be seen, when diffusion is fast (small $\tau_D$), fidelity converges to its equilibrium value and proofreading is lost

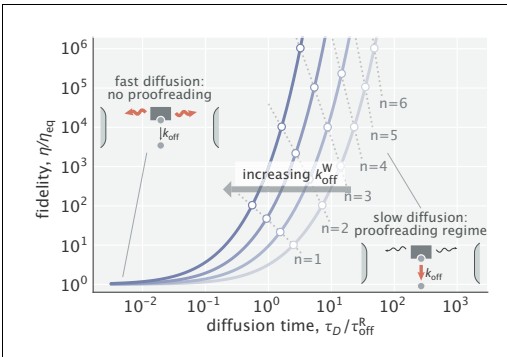

**Figure 2.** Dependence of fidelity on the diffusion time scale in the limit of very high substrate localization. Individual curves were made for different choices of $k_{off}^W$ (varied in the $[10 - 100]\,k_{off}^R$ range). $\tau_{off}^R = 1/k_{off}^R$ is the unbinding time scale of right substrates, kept fixed in the study. Fidelity values corresponding to integer degrees of proofreading in a traditional sense ($\eta/\eta_{eq} = \eta_{eq}^n$, $n = 1, 2, 3, ...$) are marked as circles. Dominant processes in the two limiting regimes are highlighted in red in the schematics shown as insets.

$(\eta \approx \sqrt{\eta_{eq}} \times \sqrt{\tau_D k_{off}^W / \tau_D k_{off}^R} = \eta_{eq})$. Conversely, when diffusion is slow (large $\tau_D$), the enzyme undergoes multiple rounds of binding a substrate at the left end and unbinding midway until it manages to diffuse across the whole compartment as a complex and form a product. These rounds serve as 'futile cycles' that endow the system with proofreading. In this regime, fidelity scales as

$$\eta \sim e^{\left(\sqrt{k_{off}^W} - \sqrt{k_{off}^R}\right)\sqrt{\tau_D}}. \tag{6}$$

To get further insights, we introduce an effective number of extra biochemical intermediates ($n$) that a traditional proofreading scheme would need to have in order to yield the same fidelity, that is $\eta/\eta_{eq} = \eta_{eq}^n$. We calculate this number as (see Appendix 1)

$$n \approx \frac{\sqrt{\tau_D k_{off}^W}}{\ln \eta_{eq}}. \tag{7}$$

Notably, since $\tau_D \sim L^2$, the result above suggests a linear relationship between the effective number of proofreading realizations and the compartment size ($n \sim L$). In addition, because the right-hand side of *Equation 7* is an increasing function of $k_{off}^W$, the proofreading efficiency of the scheme rises with larger differences in substrate off-rates (*Figure 2*) – a feature that 'hard–wired' traditional proofreading schemes with a fixed number of proofreading steps lack.

## Navigating the speed–fidelity trade-off

As is inherent to all proofreading schemes, the fidelity enhancement described earlier comes at a cost of reduced product formation speed. This reduction, in our case, happens because of increased delays in diffusive transport. Here, we explore the resulting speed–fidelity trade-off and its different regimes by varying two of the model parameters: diffusion time scale $\tau_D$ and the substrate localization length scale $\lambda_s$.

Speed and fidelity for different sampled values of $\tau_D$ and $\lambda_s$ are depicted in *Figure 3a*. As can be seen, for a fixed $\tau_D$, the reduction of $\lambda_s$ can trade off fidelity against speed. This trade-off is intuitive; with tighter substrate localization, the complexes are formed closer to the left boundary. Hence, a smaller fraction of complexes reach the activation region, reducing reaction speed. The Pareto-optimal front of the trade-off over the whole parameter space, shown as a red curve on the plot, is reached in the limit of ideal substrate localization ($\lambda_s \to 0$). Varying the diffusion time scale allows one to navigate this optimal trade-off curve and access different performance regimes.

Specifically, if the diffusion time scale is fast compared with the time scales of substrate unbinding (i.e. $\tau_D \ll 1/k_{off}^R, 1/k_{off}^W$), then both right and wrong complexes that form near the left boundary arrive at the activation region with high probability, resulting in high speeds, although at the expense of error–prone product formation (*Figure 3b*, top). In the opposite limit of slow diffusion, both types of complexes have exponentially low densities at the activation region, but due to the difference in substrate off-rates, production is highly accurate (*Figure 3b*, bottom). There also exists an intermediate regime where a significant fraction of right complexes reach the activation region while the vast majority of wrong complexes do not (*Figure 3b*, middle). As a result, an advantageous trade-off is achieved where a moderate decrease in the production rate yields high fidelity enhancement – a feature that was also identified in multi-step traditional proofreading models (*Murugan et al., 2012*).

In Appendix 3, we also study this trade-off caused by varying the catalysis rate $r$. Briefly, we find that when all other parameters are fixed, increasing $r$ trades off fidelity against speed in a linear fashion, with the ratio of highest and lowest fidelity values falling in the $[\sqrt{\eta_{eq}}, \eta_{eq}]$ range. The Pareto–

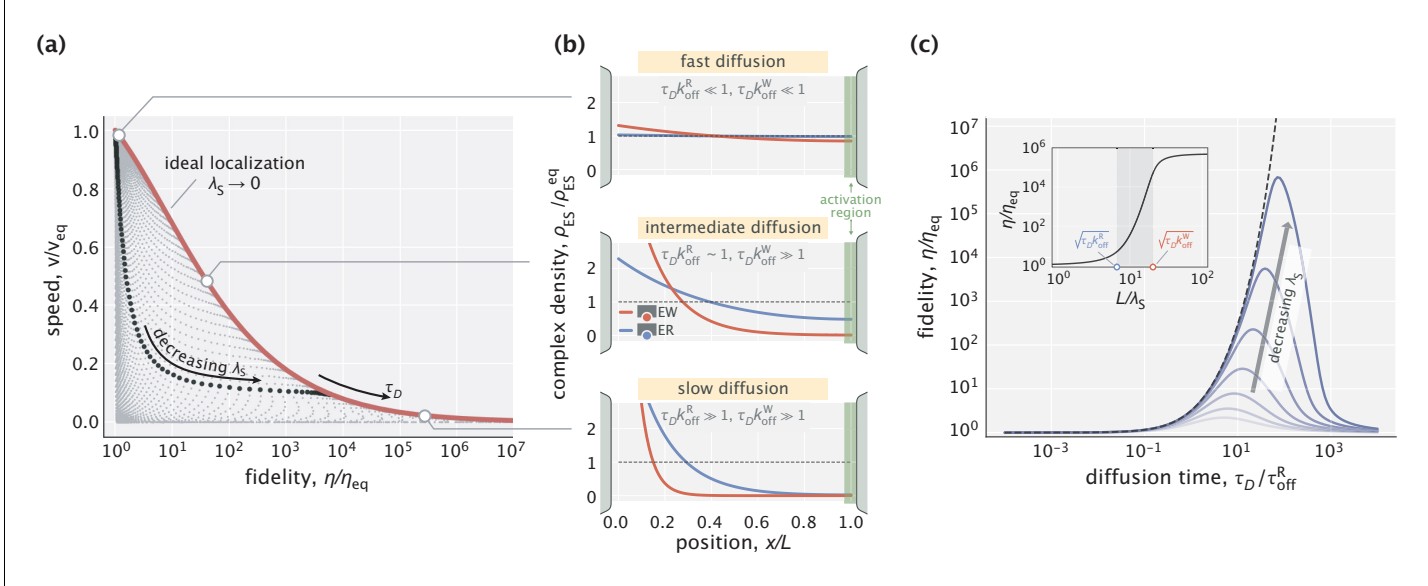

**Figure 3.** Speed–fidelity trade-off and consequences of having weak substrate gradients. (a) Speed and fidelity evaluated for sampled values of the diffusion time scale ($\tau_D$) and substrate localization length scale ($\lambda_s$). Here, $v_{eq} \sim 1/k_{off}^R$ is the speed in the equilibrium limit of a uniform substrate profile ($\lambda_s \to \infty$). The red line corresponds to the Pareto-optimal front and is reached in the high substrate localization limit. The example speed–fidelity trade-off illustrated through the black dotted curve is obtained for $\tau_D \approx 20\,\tau_{off}^R$. (b) Density profiles of wrong (EW) and right (ER) complexes in three qualitatively different performance regimes. The normalization factor $\rho_{ES}^{eq}$ corresponds to the equilibrium complex densities. (c) Fidelity as a function of diffusion time scale for different choices of $\lambda_s$ (varied in the $[0.04, 0.4]\,L$ range). The dashed line corresponds to the ideal substrate localization limit ($\lambda_s \to 0$). Inset: Fidelity as a function of $L/\lambda_s$ for a fixed $\tau_D$. Shaded area indicates the range where the bulk of fidelity enhancement takes place. Equilibrium fidelity $\eta_{eq} = 10$ was used in generating all the panels.

optimal front of the trade-off, however, monotonically shifts toward the higher speed region, suggesting that faster catalysis is, in fact, more favorable if the diffusion time scale $\tau_D$ can be adjusted accordingly (see Appendix 3 for details).

We saw in *Figure 3a* that in the case of ideal substrate localization, the slowdown of diffusive transport necessarily reduced the production rate and increased the fidelity. The latter part of this statement, however, breaks down when substrate gradients are weak. Indeed, fidelity exhibits a non-monotonic response to tuning $\tau_D$ when the substrate gradient length scale $\lambda_s$ is non-zero (*Figure 3c*). The reason for the eventual decay in fidelity is the fact that with slower diffusion (larger $\tau_D$), substrate binding and unbinding events take place more locally and therefore, the right and wrong complex profiles start to resemble the substrate profile itself, which does not discriminate between the two substrate kinds. We show in Appendix 1 that the optimal diffusion time scale can be roughly approximated as $\tau_D^*/\tau_{off}^R \approx \eta_{eq}^{-1}(L/\lambda_s)^2$, which increases monotonically with $L/\lambda_s$, consistent with the shifting peaks in *Figure 3c*.

Not surprisingly, the error–correcting capacity of the scheme improves with better substrate localization (lower $\lambda_s$). For a fixed $\tau_D$, the bulk of this improvement takes place when $L/\lambda_s$ is tuned in a range set by the two key dimensionless numbers of the model, namely, $\sqrt{\tau_D k_{off}^R}$ and $\sqrt{\tau_D k_{off}^W}$ (*Figure 3c*, inset). In Appendix 1, we provide an analytical justification for this result. Taken together, these parametric studies uncover the operational principles of the spatial proofreading scheme and demonstrate how the speed–fidelity trade-off could be dynamically navigated as needed by tuning the key time and length scales of the model.

## Energy dissipation and limits of proofreading performance

A hallmark signature of proofreading is that it is a nonequilibrium mechanism with an associated free energy cost. In our scheme, the enzyme itself is not directly involved in any energy-consuming reactions, such as hydrolysis. Instead, the free energy cost comes from maintaining the spatial gradient of substrates, which the enzymatic reaction tends to homogenize by releasing bound substrates in

regions of low substrate concentration. As the activating effectors are assumed to be tethered at $x = L$, they do not require dissipation to remain localized.

While mechanisms of substrate gradient maintenance may differ in their energetic efficiency, there exists a thermodynamically dictated minimum energy that any such mechanism must dissipate per unit time. We calculate this minimum power $P$ as

$$P = \sum_{S=\{R,W\}} \int_0^L j_S(x)\mu(x)\,dx. \qquad (8)$$

Here $j_S(x) = k_{on}\rho_S(x)\rho_E - k_{off}^S\rho_{ES}(x)$ is the net local binding flux of substrate 'S', and $\mu(x)$ is the local chemical potential (see Appendix 2.1 for details). For substrates with an exponentially decaying profile considered here, the chemical potential is given by

$$\mu(x) = \mu(0) + k_BT\ln\frac{\rho_S(x)}{\rho_S(0)} = \mu(0) - k_BT\frac{x}{\lambda_S}, \qquad (9)$$

where $k_BT$ is the thermal energy scale. Notably, the chemical potential difference across the compartment, which serves as an effective driving force for the scheme, is set by the inverse of the non-dimensionalized substrate localization length scale, namely,

$$\beta\Delta\mu = \frac{L}{\lambda_S}, \qquad (10)$$

where $\beta^{-1} = k_BT$. This driving force is zero for a uniform substrate profile ($\lambda_S \to \infty$) and increases with tighter localization (lower $\lambda_S$), as intuitively expected.

We used *Equation 8* to study the relationship between dissipation and fidelity enhancement as we tuned $\Delta\mu$ for different choices of the diffusion time scale $\tau_D$. As can be seen in *Figure 4*, power rises with increasing fidelity, diverging when fidelity reaches its asymptotic maximum given by *Equation 5* in the large $\Delta\mu$ limit. For the bulk of each curve, power scales as the logarithm of fidelity, suggesting that a linear increase in dissipation can yield an exponential reduction in error. Notably, such a scaling relationship has also been calculated in the context of *E. coli* chemoreceptor adaptation (*Lan et al., 2012*). In particular, it was shown that the adaptation error decreases exponentially with energy dissipated through multiple methylation–demethylation cycles which are used to stabilize the activity state of the receptor. Analogies in the cost-performance trade-off across these functionally distinct mechanisms contribute to the search for overarching thermodynamic themes underlying cellular information processing (*Lan et al., 2012*; *Lan and Tu, 2013*; *Horowitz et al., 2017*; *Sartori and Pigolotti, 2015*).

The logarithmic scaling is achieved in our model when the driving force is in a range where most of the fidelity enhancement takes place, namely,

$$\beta\Delta\mu \in \left[\sqrt{\tau_D k_{off}^R}, \sqrt{\tau_D k_{off}^W}\right]. \qquad (11)$$

At the end of this range, the cost per substrate binding event approaches $\sqrt{\eta_{eq}}$ in $k_BT$ units (see Appendix 2.1 for details). And beyond the range, additional error correction is attained at an increasingly higher cost.

Note that the power computed here does not include the baseline cost of creating the substrate gradient, which, for instance, would depend on the substrate diffusion constant. We

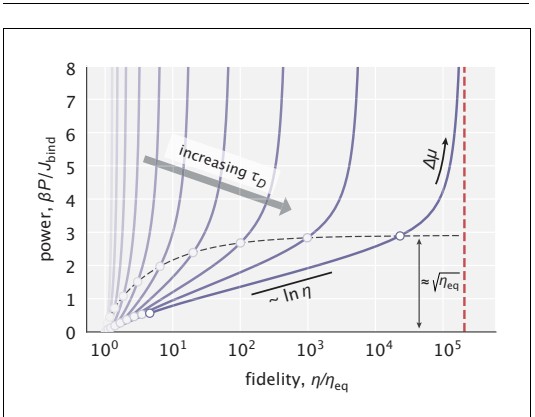

**Figure 4.** Power–fidelity relationship when tuning the effective driving force $\Delta\mu$ for different choices of the diffusion time scale $\tau_D$. $J_{bind} = k_{on}\rho_E\int\rho_S(x)\,dx$ is the integrated rate of substrate binding. The red line indicates the large dissipation limit of fidelity given by *Equation 5*. The circles indicate the $\Delta\mu$ range specified in *Equation 11* for different $\tau_D$ choices. For sufficiently large $\tau_D$ values, the cost per binding event approaches $\beta\sqrt{\eta_{eq}}$ at the end of this range (see Appendix 2.1 for details). In making this plot, $\eta_{eq} = 10$ was used.

only account for the additional cost to be paid due to the operation of the proofreading scheme which works to homogenize this substrate gradient. The baseline cost in our case is analogous to the work that ATP synthase needs to perform to maintain a nonequilibrium [ATP]/[ADP] ratio in the cell, whereas our calculated power is analogous to the rate of ATP hydrolysis by a traditional proofreading enzyme. We discuss these two classes of dissipation in greater detail in Appendix 2.3.

Just as the cellular chemical potential of ATP or GTP imposes a thermodynamic upper bound on the fidelity enhancement by any proofreading mechanism (**Qian, 2006**), the effective driving force $\Delta\mu$ imposes a similar constraint for the spatial proofreading model. This thermodynamic limit depends only on the available chemical potential and is equal to $e^{\beta\Delta\mu}$. This limit can be approached very closely by our model, which for $\Delta\mu \geq 1$ achieves the exponential enhancement with an additional linear prefactor, namely, $(\eta/\eta_{\mathrm{eq}})^{\mathrm{max}} \approx e^{\beta\Delta\mu}/\beta\Delta\mu$ (see Appendix 2.2). Such scaling behavior was theoretically accessible only to infinite-state traditional proofreading schemes (**Qian, 2006**; **Ehrenberg and Blomberg, 1980**). This offers a view of spatial proofreading as a procession of the enzyme through an infinite series of spatial filters and suggests that, from the perspective of peak error reduction capacity, our model outperforms the finite-state schemes.

## Proofreading by biochemically plausible intracellular gradients

Our discussion of the minimal model thus far was not aimed at a particular biochemical system and thus did not involve the use of realistic reaction rates and diffusion constants typically seen in living cells. Furthermore, we did not account for the possibility of substrate diffusion, as well as for the homogenization of substrate concentration gradients due to enzymatic reactions, and have thereby abstracted away the gradient maintaining mechanism. The quantitative inspection of such mechanisms is important for understanding the constraints on spatial proofreading in realistic settings.

Here, we investigate proofreading based on a widely applicable mechanism for creating gradients by the spatial separation of two opposing enzymes (**Stelling and Kholodenko, 2009**; **Bivona et al., 2003**; **Brown and Kholodenko, 1999**). Consider a protein $S$ that in its free state is phosphorylated by a membrane-bound kinase and dephosphorylated by a delocalized cytoplasmic phosphatase, as shown in **Figure 5a**. This setup will naturally create a gradient of the active form of protein ($S^*$), with the gradient length scale controlled by the rate of phosphatase activity $k_{\mathrm{p}}$ ($S^* \xrightarrow{k_{\mathrm{p}}} S$). Such mechanisms are known to create gradients of the active forms of MEK and ERK (**Kholodenko, 2006**), of GTPases such as Ran (with GEF and GAP [**Kalab et al., 2002**] playing the role of kinase and phosphatase, respectively), of cAMP (**Kholodenko, 2006**) and of stathmin oncoprotein 18 (Op18) (**Bastiaens et al., 2006**; **Niethammer et al., 2004**) near the plasma membrane, the Golgi apparatus, the ER, kinetochores and other places.

We test the proofreading power of such gradients, assuming experimentally constrained biophysical parameters for the gradient forming mechanism. Specifically, we consider an enzyme $E$ that acts on the active forms of cognate ($R^*$) and non-cognate ($W^*$) substrates which have off-rates 0.1 s$^{-1}$ and 1 s$^{-1}$, respectively (hence, $\eta_{\mathrm{eq}} = 10$). These off-rates are consistent with typical values for substrates proofread by cellular signaling systems (**Cui and Mehta, 2018**; **Gascoigne et al., 2001**). The kinases and phosphatases in our setup act identically on right and wrong substrates. We consider a dephosphorylation rate constant $k_{\mathrm{p}} = 5$ s$^{-1}$ that falls in the range $0.1-100$ s$^{-1}$ reported for different phosphatases (**Brown and Kholodenko, 1999**; **Kholodenko et al., 2000**; **Todd et al., 1999**), and a cytosolic diffusion constant $D = 1$ μm$^2$/s for all proteins in this model. With this setup, exponential gradients of length scale ~0.5 μm are formed for $R^*$ and $W^*$. We evaluate the proofreading and energetic performance of the model in a compartment of size $L = 10$ μm – a typical length scale in eukaryotic cells (see Appendix 6 for details).

Although not cost-efficient, this setup achieves proofreading in a wide range of regimes. Specifically, it is most effective when the enzyme–substrate binding is slow, in which case the exponential substrate profile is maintained and the system attains the fidelity predicted by our earlier explanatory model (**Figure 5b**). The system's proofreading capacity is retained if the first–order on-rate is raised up to $k_{\mathrm{on}}\rho_{\mathrm{E}} \sim 10$ s$^{-1}$, where around 10-fold increase in fidelity is still possible. If the binding rate constant ($k_{\mathrm{on}}$) or the enzyme's expression level ($\rho_{\mathrm{E}}$) is any higher, then enzymatic reactions overwhelm the ability of the kinase/phosphatase system to keep the active forms of substrates sufficiently localized (**Figure 5c**) and proofreading is lost. Overall, this model suggests that enzymes can work at

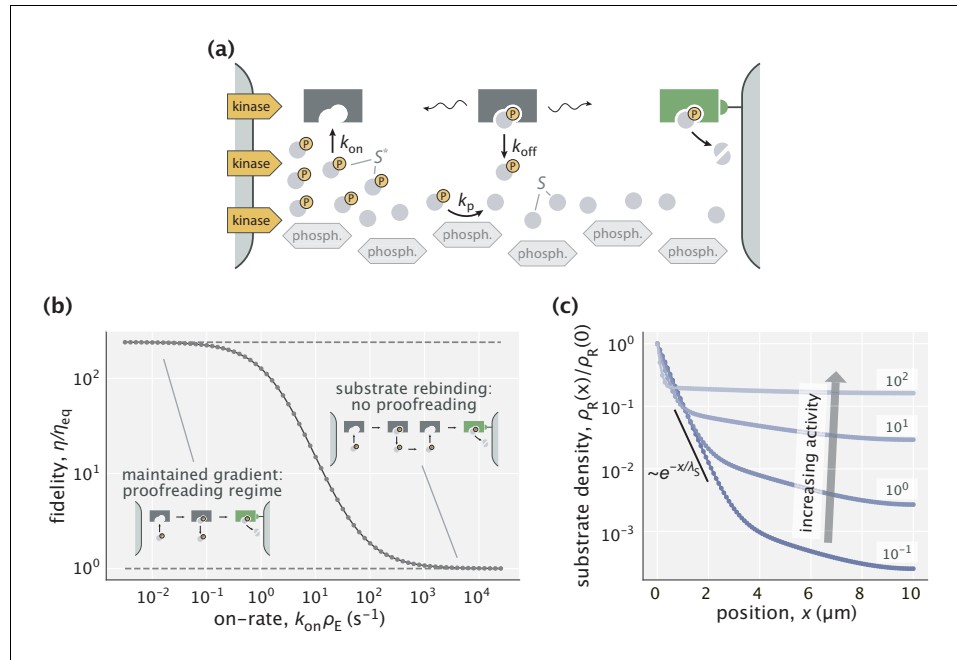

**Figure 5.** Proofreading based on substrate gradients formed by spatially separated kinases and phosphatases. (a) The active form $S^*$ of many proteins exhibits gradients because kinases that phosphorylate $S$ are anchored to a membrane while phosphatases can diffuse in the cytoplasm (*Kholodenko, 2006*). An enzyme can exploit the resulting spatial gradient for proofreading. (b) At low enzyme activity (i.e. low $k_{on}\rho_E$), the gradient of $S^*$ is successfully maintained, allowing for proofreading. The upper dashed line corresponds to the peak fidelity when the substrate profile is exponential. At high enzyme activity (large $k_{on}\rho_E$), the dephosphorylation with rate $k_p = 5$ s$^{-1}$ is no longer sufficient to maintain the gradient and proofreading is lost. (c) Profiles of right substrates for different choices of enzyme activity. Numbers indicate $k_{on}\rho_E$ in s$^{-1}$ units. The black line shows an exponential substrate profile with a length scale $\lambda_s = \sqrt{D/k_p} \sim 0.5$ μm.

reasonable binding rates and still proofread, when accounting for an experimentally characterized gradient maintaining mechanism.

## Discussion

We have outlined a way for enzymatic reactions to proofread and improve specificity by exploiting spatial concentration gradients of substrates. Like the classic model, our proposed spatial proofreading scheme is based on a time delay; but unlike the classic model, here the delay is due to spatial transport rather than transitions through biochemical intermediates. Consequently, the enzyme is liberated from the stringent structural requirements imposed by traditional proofreading, such as multiple intermediate conformations and hydrolysis sites for energy coupling. Instead, our scheme exploits the free energy supplied by active mechanisms that maintain spatial structures.

The decoupling of the two crucial features of proofreading – time delay and free energy dissipation – allows the cell to tune proofreading on the fly. For instance, all proofreading schemes offer fidelity at the expense of reaction speed and energy. For traditional schemes, navigating this trade-off is not always feasible, as it needs to involve structural changes via mutations or modulation of the [ATP]/[ADP] ratio which can cause collateral effects on the rest of the cell. In contrast, the spatial proofreading scheme is more adaptable to the changing conditions and needs of the cell. The scheme can prioritize speed in one context, and fidelity in another, simply by tuning the length scale of intracellular gradients (e.g. through the regulation of the phosphotase or free enzyme concentration in the scheme discussed earlier).

On the other hand, this modular decoupling can complicate the experimental identification of proofreading enzymes and the interpretation of their fidelity. Here, the enzymes need not be endowed with the structural and biochemical properties typically sought for in a proofreading

enzyme. At the same time, any attempt to reconstitute enzymatic activity in a well-mixed, in vitro assay, will show poor fidelity compared to in vivo measurements, even when all necessary molecular players are present in vitro. Therefore, more care is required in studies of cellular information processing mechanisms that hijack a distant source of free energy compared to the case where the relevant energy consumption is local and easier to link causally to function.

While we focused on spatially localized substrates and delocalized enzymes, our framework would apply equally well to other scenarios, like one with a spatially localized enzyme (or its active form [*Kalab et al., 2002*; *Nalbant et al., 2004*]) and effector with delocalized substrates, an example of which would be an alternative version of the scheme in *Figure 5a* where the target of the kinase/phosphatase activity is changed from substrates to enzymes. Our framework can also be extended to signaling cascades, where slightly different phosphatase activities can result in magnified concentration ratios of two competing signaling molecules at the spatial location of the next cascade step (*Roy and Cyert, 2009*; *Bauman and Scott, 2002*; *Kholodenko, 2006*).

The spatial gradients needed for the operation of our model can be created and maintained through multiple mechanisms in the cell, ranging from the kinase/phosphatase system modeled here, to the passive diffusion of substrates/ligands combined with active degradation (e.g. Bicoid and other developmental morphogens), to active transport processes combined with diffusion. A particularly simple implementation of our scheme is via compartmentalization – substrates and effectors are localized in two spatially separated compartments with the enzyme–substrate complex having to travel from one to another to complete the reaction.

Many molecular localization pathways involving the naturally compartmentalized parts of the cell require high substrate selectivity and are therefore potential candidates for the implementation of spatial proofreading. For example, in polarized, asymmetric cells (e.g. budding yeast or neuronal cells) gene expression often needs to be spatially regulated (*Parton et al., 2014*; *Martin and Ephrussi, 2009*). Such regulation is achieved with designated ribonucleoproteins that bind specific mRNAs near the cell nucleus, perform a biased random walk to the mRNA localization site and deliver them for translation. During transport, mRNAs are protected from ribosome binding and when they unbind, they are subject to degradation which would prevent rebinding events at intermediate locations. Another example process is the non-vesicular transport of lipids between the membrane–bound domains of the cells (e.g. the ER, mitochondria, the Golgi apparatus, or the plasma membrane). This transport mechanism is mediated by lipid-transfer proteins that bind lipids on the donor membrane, diffuse to the acceptor membrane and upon interacting with it, undergo a conformational change, delivering the 'cargo' (*Lev, 2010*). Although the higher proximity of the two membranes is thought to enhance the transport efficiency, it would be interesting to study the optimality of the inter-membrane distance in the context of fidelity–transport efficiency trade-off, given the fact that some of the lipid-transfer proteins are known to exhibit specificity for their cognate substrates.

Our scheme may also be applicable as a quality control mechanism in protein secretion pathways (*Ellgaard and Helenius, 2003*; *Arvan et al., 2002*), in high-fidelity targeting of membrane proteins mediated by signal recognition particles (*Rao et al., 2016*; *Chio et al., 2017*), as well as in selective glycosylation reactions in the Golgi apparatus (*Jaiman and Thattai, 2020*). Lastly, considering the recent advances in generating synthetic morphogen patterns in multicellular organisms (*Toda et al., 2020*; *Stapornwongkul et al., 2020*), spatial proofreading could also be employed in pathways acting on engineered protein gradients. Experimental investigations of these processes in light of our work will reveal the extent to which spatial transport promotes specificity.

In conclusion, we have analyzed the role played by spatial structures in endowing enzymatic reactions with kinetic proofreading. Simply by spatially segregating substrate binding from catalysis, enzymes can enhance their specificity. This suggests that enzymatic reactions may acquire *de novo* proofreading capabilities by coupling to pre-existing spatial gradients in the cell.

## Materials and methods

Detailed derivations of the analytical results presented in the main text along with additional studies on our model are included in the Appendices. In addition, Python scripts and Jupyter notebooks used to generate all the plots in the main text and Appendices are included as Supplementary files.

## Acknowledgements

We thank Anatoly Kolomeisky, Shu-ou Shan and Erik Winfree for insightful discussions, Soichi Hirokawa and Avi Flamholz for providing useful feedback on the manuscript. We also thank Alexander Grosberg whose idea of a compartmentalized 'rotary demon' motivated the development of our model. This work was supported by the NIH Grant 1R35 GM118043-01, the John Templeton Foundation Grants 51250 and 60973 (to RP), a James S. McDonnell Foundation postdoctoral fellowship (to KH), and the Simons Foundation (AM).

## Additional information

### Funding

| Funder | Author |
|---|---|
| James S. McDonnell Foundation | Kabir Husain |
| Simons Foundation | Arvind Murugan |
| John Templeton Foundation | Rob Phillips |
| National Institute of General Medical Sciences | Rob Phillips |

The funders had no role in study design, data collection and interpretation, or the decision to submit the work for publication.

### Author contributions

Vahe Galstyan, Kabir Husain, Conceptualization, Formal analysis, Investigation, Methodology, Writing - original draft, Writing - review and editing; Fangzhou Xiao, Conceptualization, Formal analysis, Investigation, Methodology, Writing - review and editing; Arvind Murugan, Rob Phillips, Conceptualization, Supervision, Funding acquisition, Investigation, Methodology, Writing - original draft, Project administration, Writing - review and editing

### Author ORCIDs

Vahe Galstyan (ID) https://orcid.org/0000-0001-7073-9175
Arvind Murugan (ID) https://orcid.org/0000-0001-5464-917X
Rob Phillips (ID) https://orcid.org/0000-0003-3082-2809

### Decision letter and Author response

Decision letter https://doi.org/10.7554/eLife.60415.sa1
Author response https://doi.org/10.7554/eLife.60415.sa2

## Additional files

### Supplementary files

• Source code 1. Code files to reproduce the figures.

### Data availability

All scripts used to generate the data for making the plots are provided in supporting files.

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

# Appendix 1

## Analytical calculations of the complex density profile and fidelity

We begin this section by deriving an analytical expression for the density profile of substrate-bound enzymes ($\rho_{ES}(x)$) in the case where the $\rho(x) \approx \text{constant}$ assumption holds. Based on this result, we then obtain expressions for fidelity in low, high, and intermediate substrate localization regimes. We reserve the studies of speed and fidelity in the general case of a nonuniform free enzyme profile to Appendix 5.

## 1. Derivation of the complex density profile $\rho_{ES}(x)$

The ordinary differential equation (ODE) that defines the steady state profile of substrate-bound enzymes is

$$\underbrace{D\frac{\mathrm{d}^2\rho_{ES}}{\mathrm{d}x^2}}_{\text{diffusion}} - \underbrace{k_{\text{off}}^S \rho_{ES}(x)}_{\text{unbinding}} + \underbrace{k_{\text{on}}\rho_S(0)e^{-x/\lambda_S}\rho_E(x)}_{\text{binding}} = 0. \tag{S1}$$

Here, $\rho_S(0)$ is the substrate density at the leftmost boundary, whose value can be calculated from the condition that the total number of free substrates is $S_{\text{total}}$, namely,

$$\begin{aligned} S_{\text{total}} &= \int_{x=0}^{L} \rho_S(0)e^{-x/\lambda_S}\,\mathrm{d}x \\ &= \rho_S(0)\lambda_S\left(1 - e^{-L/\lambda_S}\right) \Rightarrow \end{aligned} \tag{S2}$$

$$\rho_S(0) = \frac{S_{\text{total}}}{\lambda_S(1 - e^{-L/\lambda_S})}. \tag{S3}$$

In the limit of low substrate amounts where the approximation $\rho_E(x) \approx \text{constant}$ is valid, *Equation S1* represents a linear nonhomogeneous ODE. Hence, its solution can be written as

$$\rho_{ES}(x) = \rho_{ES}^{(h)}(x) + \rho_{ES}^{(p)}(x), \tag{S4}$$

where $\rho_{ES}^{(h)}(x)$ is the general solution to the corresponding homogeneous equation, while $\rho_{ES}^{(p)}(x)$ is a particular solution.

Looking for solutions of the form $Ce^{-x/\lambda}$ for the homogeneous part, we find

$$C\left(\frac{D}{\lambda^2} - k_{\text{off}}^S\right)e^{-x/\lambda} = 0. \tag{S5}$$

The two possible roots for $\lambda$ are $\pm\sqrt{D/k_{\text{off}}^S}$. Calling the positive root $\lambda_{ES}$, which represents the mean distance traveled by the substrate–bound enzyme before releasing the substrate, we can write the general solution to the homogeneous part of *Equation S1* as

$$\rho_{ES}^{(h)}(x) = C_1 e^{-x/\lambda_{ES}} + C_2 e^{x/\lambda_{ES}}, \tag{S6}$$

where $C_1$ and $C_2$ are constants which will be determined from the boundary conditions.

Since the nonhomogeneous part of *Equation S1* is a scaled exponential, we look for a particular solution of the same functional form, namely, $\rho_{ES}^{(p)}(x) = C_p e^{-x/\lambda_S}$. Substituting this form into the ODE, we obtain

$$C_p\left(\frac{D}{\lambda_S^2} - k_{\text{off}}^S\right)e^{-x/\lambda_S} = -k_{\text{on}}\rho_S(0)e^{-x/\lambda_S}\rho_E. \tag{S7}$$

The constant coefficient $C_p$ can then be found as

$$C_{\mathrm{p}} = \frac{k_{\mathrm{on}}\rho_{\mathrm{S}}(0)\rho_{\mathrm{E}}}{k_{\mathrm{off}}^{\mathrm{S}} - \frac{D}{\lambda_{\mathrm{S}}^2}} = \frac{k_{\mathrm{on}}\rho_{\mathrm{S}}(0)\rho_{\mathrm{E}}}{k_{\mathrm{off}}^{\mathrm{S}}\left(1 - \frac{D/k_{\mathrm{off}}^{\mathrm{S}}}{\lambda_{\mathrm{S}}^2}\right)}$$

$$= \frac{k_{\mathrm{on}}\rho_{\mathrm{S}}(0)\rho_{\mathrm{E}}}{k_{\mathrm{off}}^{\mathrm{S}}\left(1 - \frac{\lambda_{\mathrm{ES}}^2}{\lambda_{\mathrm{S}}^2}\right)}, \tag{S8}$$

where we have used the equality $\lambda_{\mathrm{ES}} = \sqrt{D/k_{\mathrm{off}}^{\mathrm{S}}}$.

Now, to find the unknown coefficients $C_1$ and $C_2$, we impose the no-flux boundary conditions for the density $\rho_{\mathrm{ES}}(x)$ at the left and right boundaries of the compartment, namely,

$$\frac{\mathrm{d}\rho_{\mathrm{ES}}}{\mathrm{d}x}\Big|_{x=0} = -\frac{C_1}{\lambda_{\mathrm{ES}}} + \frac{C_2}{\lambda_{\mathrm{ES}}} - \frac{C_{\mathrm{p}}}{\lambda_{\mathrm{S}}} = 0, \tag{S9}$$

$$\frac{\mathrm{d}\rho_{\mathrm{ES}}}{\mathrm{d}x}\Big|_{x=L} = -\frac{C_1}{\lambda_{\mathrm{ES}}}e^{-\frac{L}{\lambda_{\mathrm{ES}}}} + \frac{C_2}{\lambda_{\mathrm{ES}}}e^{\frac{L}{\lambda_{\mathrm{ES}}}} - \frac{C_{\mathrm{p}}}{\lambda_{\mathrm{S}}}e^{-\frac{L}{\lambda_{\mathrm{S}}}} = 0. \tag{S10}$$

Note that we did not take into account the product formation flux at the rightmost boundary when writing *Equation S10* in order to simplify our calculations. This is justified in the limit of slow catalysis – an assumption that we make in our treatment. The above system of two equations can then be solved for $C_1$ and $C_2$, yielding

$$C_1 = -\frac{\lambda_{\mathrm{ES}}}{2\lambda_{\mathrm{S}}}\frac{e^{L/\lambda_{\mathrm{ES}}} - e^{-L/\lambda_{\mathrm{S}}}}{\sinh(L/\lambda_{\mathrm{ES}})}C_{\mathrm{p}}, \tag{S11}$$

$$C_2 = \frac{\lambda_{\mathrm{ES}}}{2\lambda_{\mathrm{S}}}\frac{e^{-L/\lambda_{\mathrm{S}}} - e^{-L/\lambda_{\mathrm{ES}}}}{\sinh(L/\lambda_{\mathrm{ES}})}C_{\mathrm{p}}. \tag{S12}$$

With the constant coefficients known, we obtain the general solution for the complex profile as

$$\begin{aligned}
\rho_{\mathrm{ES}}(x) &= C_1 e^{-x/\lambda_{\mathrm{ES}}} + C_2 e^{x/\lambda_{\mathrm{ES}}} + C_{\mathrm{p}}e^{-x/\lambda_{\mathrm{S}}} \\
&= C_{\mathrm{p}}\left(\frac{\lambda_{\mathrm{ES}}}{\lambda_{\mathrm{S}}\sinh(L/\lambda_{\mathrm{ES}})}\left[-\frac{e^{(L-x)/\lambda_{\mathrm{ES}}} + e^{(x-L)/\lambda_{\mathrm{ES}}}}{2} + \frac{e^{-x/\lambda_{\mathrm{ES}}} + e^{x/\lambda_{\mathrm{ES}}}}{2}e^{-L/\lambda_{\mathrm{S}}}\right] + e^{-x/\lambda_{\mathrm{S}}}\right) \\
&= \frac{k_{\mathrm{on}}\rho_{\mathrm{S}}(0)\rho_{\mathrm{E}}}{k_{\mathrm{off}}^{\mathrm{S}}\left(1 - \lambda_{\mathrm{ES}}^2/\lambda_{\mathrm{S}}^2\right)}\left(\frac{\lambda_{\mathrm{ES}}}{\lambda_{\mathrm{S}}\sinh(L/\lambda_{\mathrm{ES}})}\left[-\cosh\left(\frac{L-x}{\lambda_{\mathrm{ES}}}\right) + \cosh\left(\frac{x}{\lambda_{\mathrm{ES}}}\right)e^{-L/\lambda_{\mathrm{S}}}\right] + e^{-x/\lambda_{\mathrm{S}}}\right) \\
&= \frac{k_{\mathrm{on}}\rho_{\mathrm{S}}(0)\rho_{\mathrm{E}}}{k_{\mathrm{off}}^{\mathrm{S}}\left(1 - \lambda_{\mathrm{ES}}^2/\lambda_{\mathrm{S}}^2\right)}\left(\frac{\lambda_{\mathrm{ES}}}{\lambda_{\mathrm{S}}\sinh(L/\lambda_{\mathrm{ES}})}\left[-\cosh\left(\frac{L-x}{\lambda_{\mathrm{ES}}}\right) + \cosh\left(\frac{x}{\lambda_{\mathrm{ES}}}\right)e^{-L/\lambda_{\mathrm{S}}}\right] + e^{-x/\lambda_{\mathrm{S}}}\right).
\end{aligned} \tag{S13}$$

## 2. Density profile in low and high substrate localization regimes

If substrate localization is very poor $(\lambda_{\mathrm{S}} \gg L)$, the substrate distribution will be uniform $(\rho_{\mathrm{S}}(x) = \bar{\rho}_{\mathrm{S}} = S_{\mathrm{total}}/L)$, resulting in a similarly flat profile of enzyme–substrate complexes with their density $\rho_{\mathrm{ES}}^{\infty}$ given by

$$\begin{aligned}
\rho_{\mathrm{ES}}^{\infty} &= \frac{k_{\mathrm{on}}\rho_{\mathrm{S}}(0)\rho_{\mathrm{E}}}{k_{\mathrm{off}}^{\mathrm{S}}} \\
&= \frac{k_{\mathrm{on}}\bar{\rho}_{\mathrm{S}}\rho_{\mathrm{E}}}{k_{\mathrm{off}}^{\mathrm{S}}}.
\end{aligned} \tag{S14}$$

This is the expected equilibrium result where the complex concentration is inversely proportional to the dissociation constant $(k_{\mathrm{off}}^{\mathrm{S}}/k_{\mathrm{on}})$.

In the opposite limit where the substrates are highly localized $(\lambda_{\mathrm{S}} \ll \lambda_{\mathrm{ES}}, L$ and $\rho_{\mathrm{S}}(0) \approx S_{\mathrm{total}}/\lambda_{\mathrm{S}}$ from *Equation S3*), the complex density profile simplifies into

$$\begin{aligned}
\rho_{ES}(x) &\approx \frac{k_{on}S_{total}\rho_E}{k_{off}^S\lambda_S(-\lambda_{ES}^2/\lambda_S^2)}\left(-\frac{\lambda_{ES}}{\lambda_S\sinh(L/\lambda_{ES})}\cosh\left(\frac{L-x}{\lambda_{ES}}\right)\right)\\
&= \frac{k_{on}S_{total}\rho_E}{k_{off}^S L}\frac{L/\lambda_{ES}}{\sinh(L/\lambda_{ES})}\cosh\left(\frac{L-x}{\lambda_{ES}}\right)\\
&= \rho_{ES}^\infty \times \frac{L/\lambda_{ES}}{\sinh(L/\lambda_{ES})}\cosh\left(\frac{L-x}{\lambda_{ES}}\right).
\end{aligned} \tag{S15}$$

The $x$-dependence through the $\cosh(\cdot)$ function suggests that the complex density is the highest at the leftmost boundary and lowest at the rightmost boundary, with the degree of complex localization dictated by the length scale parameter $\lambda_{ES}$. Notably, this localization of complexes does not alter their total number, since the average complex density is conserved, that is,

$$\begin{aligned}
\langle\rho_{ES}\rangle &= \int_0^L \rho_{ES}(x)\,dx\\
&= \rho_{ES}^\infty \times \frac{L/\lambda_{ES}}{\sinh(L/\lambda_{ES})}\times\frac{1}{L}\int_0^L\cosh\left(\frac{L-x}{\lambda_{ES}}\right)dx\\
&= \rho_{ES}^\infty \times \frac{L/\lambda_{ES}}{\sinh(L/\lambda_{ES})}\times\frac{\lambda_{ES}}{L}\sinh(L/\lambda_{ES})\\
&= \rho_{ES}^\infty.
\end{aligned} \tag{S16}$$

*Equation S15* for the complex profile can be alternatively written in terms of the diffusion time scale $\tau_D = L^2/D$ and the substrate off-rate $k_{off}^S$. Noting that $L/\lambda_{ES} = \sqrt{L^2 k_{off}^S/D} = \sqrt{\tau_D k_{off}^S}$ and introducing a dimensionless coordinate $\tilde{x} = x/L$, we find

$$\rho_{ES}(x) = \rho_{ES}^\infty \times \frac{\sqrt{\tau_D k_{off}^S}}{\sinh\left(\sqrt{\tau_D k_{off}^S}\right)}\cosh\left(\sqrt{\tau_D k_{off}^S}(1-\tilde{x})\right). \tag{S17}$$

The above equation is what was used for generating the plots in *Figure 3b* of the main text for different choices of the diffusion time scale.

## 3. Fidelity in low and high substrate localization regimes

Let us now evaluate the fidelity of the model in the two limiting regimes discussed earlier. In the poor substrate localization case, which corresponds to an equilibrium setting, the fidelity can be found from *Equation S14* as

$$\eta_{eq} = \frac{r\rho_{ER}^\infty}{r\rho_{EW}^\infty} = \frac{k_{off}^W}{k_{off}^R}, \tag{S18}$$

where we have employed the assumption about the right and wrong substrates having identical density profiles. This is the expected result for equilibrium discrimination where no advantage is taken of the system's spatial structure.

In the regime with high substrate localization, the enzyme–substrate complexes have a nonuniform spatial distribution. What matters for product formation is the complex density at the rightmost boundary ($\tilde{x} = 1$), which we obtain from *Equation S17* as

$$\rho_{ES}(L) = \rho_{ES}^\infty \times \frac{\sqrt{\tau_D k_{off}^S}}{\sinh\left(\sqrt{\tau_D k_{off}^S}\right)}. \tag{S19}$$

Substituting the above expression written for right and wrong complexes into the definition of fidelity, we find

$$\begin{aligned}
\eta &= \frac{r\rho_{\mathrm{ER}}(L)}{r\rho_{\mathrm{EW}}(L)} \\
&= \eta_{\mathrm{eq}} \times \sqrt{\frac{k_{\mathrm{off}}^{\mathrm{R}}}{k_{\mathrm{off}}^{\mathrm{W}}}} \frac{\sinh\left(\sqrt{\tau_D k_{\mathrm{off}}^{\mathrm{W}}}\right)}{\sinh\left(\sqrt{\tau_D k_{\mathrm{off}}^{\mathrm{R}}}\right)} \\
&= \sqrt{\eta_{\mathrm{eq}}} \frac{\sinh\left(\sqrt{\tau_D k_{\mathrm{off}}^{\mathrm{W}}}\right)}{\sinh\left(\sqrt{\tau_D k_{\mathrm{off}}^{\mathrm{R}}}\right)}.
\end{aligned} \tag{S20}$$

This is the result reported in *Equation 5* of the main text. To gain more intuition about it and draw parallels with traditional kinetic proofreading, let us consider the limit of long diffusion time scales where proofreading is the most effective. In this limit, the hyperbolic sine functions above can be approximated as $\sinh(\sqrt{\tau_D k_{\mathrm{off}}^{\mathrm{S}}}) \approx 0.5\,e^{\sqrt{\tau_D k_{\mathrm{off}}^{\mathrm{S}}}}$, simplifying the fidelity expression into

$$\begin{aligned}
\eta &= \sqrt{\eta_{\mathrm{eq}}} \frac{e^{\sqrt{\tau_D k_{\mathrm{off}}^{\mathrm{W}}}}}{e^{\sqrt{\tau_D k_{\mathrm{off}}^{\mathrm{R}}}}} \\
&= \sqrt{\eta_{\mathrm{eq}}}\, e^{\sqrt{\tau_D k_{\mathrm{off}}^{\mathrm{W}}} - \sqrt{\tau_D k_{\mathrm{off}}^{\mathrm{R}}}} \\
&= \sqrt{\eta_{\mathrm{eq}}}\, e^{\sqrt{\tau_D k_{\mathrm{off}}^{\mathrm{R}}}\left(\sqrt{\eta_{\mathrm{eq}}} - 1\right)},
\end{aligned} \tag{S21}$$

where we have used the definition of equilibrium fidelity (*Equation S18*). In traditional proofreading, a scheme with $n$ proofreading realizations can yield a maximum fidelity of $\eta/\eta_{\mathrm{eq}} = \eta_{\mathrm{eq}}^n$. The value of $n$ for the original Hopfield model, for instance, is 1. It would be informative to also know the effective parameter $n$ for the spatial proofreading model. Dividing *Equation S21* by $\eta_{\mathrm{eq}}$, we find

$$\begin{aligned}
\frac{\eta}{\eta_{\mathrm{eq}}} = \frac{1}{\sqrt{\eta_{\mathrm{eq}}}} e^{\sqrt{\tau_D k_{\mathrm{off}}^{\mathrm{R}}}\left(\sqrt{\eta_{\mathrm{eq}}} - 1\right)} &= \eta_{\mathrm{eq}}^n, \\
e^{\sqrt{\tau_D k_{\mathrm{off}}^{\mathrm{R}}}\left(\sqrt{\eta_{\mathrm{eq}}} - 1\right)} &= \eta_{\mathrm{eq}}^{n+\frac{1}{2}}, \\
\sqrt{\tau_D k_{\mathrm{off}}^{\mathrm{R}}}\left(\sqrt{\eta_{\mathrm{eq}}} - 1\right) &= \left(n + \frac{1}{2}\right)\ln\eta_{\mathrm{eq}} \Rightarrow \\
n + \tfrac{1}{2} &= \frac{\sqrt{\eta_{\mathrm{eq}}} - 1}{\ln\eta_{\mathrm{eq}}}\sqrt{\tau_D k_{\mathrm{off}}^{\mathrm{R}}}.
\end{aligned} \tag{S22}$$

This exact result can be simplified into an approximate form when diffusion is slow and $\eta_{\mathrm{eq}} \gg 1$, yielding the expression reported in *Equation 7* of the main text, namely,

$$\begin{aligned}
n &\approx \frac{\sqrt{\eta_{\mathrm{eq}}}\sqrt{\tau_D k_{\mathrm{off}}^{\mathrm{R}}}}{\ln\eta_{\mathrm{eq}}} \\
&= \frac{\sqrt{\tau_D k_{\mathrm{off}}^{\mathrm{W}}}}{\ln\eta_{\mathrm{eq}}}.
\end{aligned} \tag{S23}$$

## 4. Fidelity in an intermediate substrate localization regime

The generic expression for complex density at the rightmost boundary ($x = L$) can be written using *Equation S13* as

$$\rho_{\mathrm{ES}}(L) = \frac{k_{\mathrm{on}}\rho_{\mathrm{S}}(0)\rho_{\mathrm{E}}}{k_{\mathrm{off}}^{\mathrm{S}}\left(1 - \lambda_{\mathrm{ES}}^2/\lambda_{\mathrm{S}}^2\right)}\left(\frac{\lambda_{\mathrm{ES}}}{\lambda_{\mathrm{S}}\sinh(L/\lambda_{\mathrm{ES}})}\left[\cosh\left(\frac{L}{\lambda_{\mathrm{ES}}}\right)e^{-L/\lambda_{\mathrm{S}}} - 1\right] + e^{-L/\lambda_{\mathrm{S}}}\right). \tag{S24}$$

For the system to proofread, substrates need to be sufficiently localized ($\lambda_{\mathrm{S}} < L$) and diffusion needs to be sufficiently slow ($\tau_D k_{\mathrm{off}}^{\mathrm{S}} > 1$ or, $\lambda_{\mathrm{ES}} < L$). Under these conditions, the substrate profile can be approximated using *Equation S3* as $\rho_{\mathrm{S}}(x) \approx \lambda_{\mathrm{S}}^{-1}S_{\mathrm{total}}e^{-x/\lambda_{\mathrm{S}}}$, while the hyperbolic sine and cosine

functions used above can be approximated as $\sinh(L/\lambda_{\mathrm{ES}}) \approx \cosh(L/\lambda_{\mathrm{ES}}) \approx 0.5\, e^{L/\lambda_{\mathrm{ES}}}$. With these approximations, the complex density expression simplifies into

$$\rho_{\mathrm{ES}}(L) = \frac{k_{\mathrm{on}} S_{\mathrm{total}} \rho_{\mathrm{E}}}{k_{\mathrm{off}}^{\mathrm{S}} \lambda_{\mathrm{S}} \left(1 - \lambda_{\mathrm{ES}}^2/\lambda_{\mathrm{S}}^2\right)} \left( \frac{\lambda_{\mathrm{ES}}}{\lambda_{\mathrm{S}}} \left[ e^{-L/\lambda_{\mathrm{S}}} - 2 e^{-L/\lambda_{\mathrm{ES}}} \right] + e^{-L/\lambda_{\mathrm{S}}} \right)$$
$$= \frac{k_{\mathrm{on}} S_{\mathrm{total}} \rho_{\mathrm{E}}}{k_{\mathrm{off}}^{\mathrm{S}} (\lambda_{\mathrm{S}}^2 - \lambda_{\mathrm{ES}}^2)} \left( (\lambda_{\mathrm{S}} + \lambda_{\mathrm{ES}}) e^{-L/\lambda_{\mathrm{S}}} - 2 \lambda_{\mathrm{ES}} e^{-L/\lambda_{\mathrm{ES}}} \right). \tag{S25}$$

Now, depending on how $\lambda_{\mathrm{S}}$ compares with $\lambda_{\mathrm{ES}}$, there can be two qualitatively different regimes for the complex density, namely,

$$\rho_{\mathrm{ES}}(L) = \rho_{\mathrm{ES}}^{\infty} \times \begin{cases} \dfrac{2L}{\lambda_{\mathrm{ES}}} e^{-L/\lambda_{\mathrm{ES}}}, & \text{if } \lambda_{\mathrm{S}} \ll \lambda_{\mathrm{ES}} \ (L/\lambda_{\mathrm{S}} \gg \sqrt{\tau_D k_{\mathrm{off}}^{\mathrm{S}}}) \\[2ex] \dfrac{L}{\lambda_{\mathrm{S}}} e^{-L/\lambda_{\mathrm{S}}}, & \text{if } \lambda_{\mathrm{ES}} \ll \lambda_{\mathrm{S}} \ (\sqrt{\tau_D k_{\mathrm{off}}^{\mathrm{S}}} \gg L/\lambda_{\mathrm{S}}) \end{cases} \tag{S26}$$

where we used the equilibrium complex density $\rho_{\mathrm{ES}}^{\infty}$ defined in *Equation S14*.

Notably, the first regime effectively corresponds to the case of ideal substrate localization where complex density is independent of the precise value of $\lambda_{\mathrm{S}}$. The dimensionless number $\sqrt{\tau_D k_{\mathrm{off}}^{\mathrm{S}}}$ sets the scale for the minimum $L/\lambda_{\mathrm{S}}$ value beyond which ideal localization can be assumed. Conversely, the second regime corresponds to the case where the distance traveled by a complex before dissociating is so short that the complex profile is dictated by the substrate profile itself. Because of that, the complex density reduction from its equilibrium limit is independent of the precise values of $\tau_D$ and $k_{\mathrm{off}}^{\mathrm{S}}$, as long as the condition $\lambda_{\mathrm{ES}} \ll \lambda_{\mathrm{S}}$ is met.

The scheme yields its highest fidelity when both right and wrong complex densities are in the first regime (ideal localization). When both densities are in the second regime, fidelity is reduced down to its equilibrium value $\eta_{\mathrm{eq}}$ (*Appendix 1—table 1*). The transition between these two extremes happens when the density profiles of right and wrong complexes fall under different regimes. Fidelity can be navigated in the transition zone by tuning the substrate gradient length scale $\lambda_{\mathrm{S}}$. This is demonstrated in *Appendix 1—figure 1* for three different choices of $\eta_{\mathrm{eq}}$. In all three cases, the dimensionless numbers $\sqrt{\tau_D k_{\mathrm{off}}^{\mathrm{R}}}$ and $\sqrt{\tau_D k_{\mathrm{off}}^{\mathrm{W}}}$ set the approximate range in which the bulk of fidelity enhancement occurs, as stated in the main text.

**Appendix 1—table 1.** Fidelity of the scheme in different regimes of right and wrong complex densities.

The upper-right cell is empty because the two conditions on $\lambda_{\mathrm{S}}$ cannot be simultaneously met, since $\lambda_{\mathrm{ER}} > \lambda_{\mathrm{EW}}$ by construction (follows from $k_{\mathrm{off}}^{\mathrm{R}} < k_{\mathrm{off}}^{\mathrm{W}}$).

| | $\lambda_{\mathrm{S}} \ll \lambda_{\mathrm{ER}}$ | $\lambda_{\mathrm{S}} \gg \lambda_{\mathrm{ER}}$ |
|---|---|---|
| $\lambda_{\mathrm{S}} \ll \lambda_{\mathrm{EW}}$ | $\dfrac{\lambda_{\mathrm{EW}}}{\lambda_{\mathrm{ER}}} e^{L\left(\lambda_{\mathrm{EW}}^{-1} - \lambda_{\mathrm{ER}}^{-1}\right)}$ | - |
| $\lambda_{\mathrm{S}} \gg \lambda_{\mathrm{EW}}$ | $\dfrac{2\lambda_{\mathrm{S}}}{\lambda_{\mathrm{ER}}} e^{L\left(\lambda_{\mathrm{S}}^{-1} - \lambda_{\mathrm{ER}}^{-1}\right)}$ | $\eta_{\mathrm{eq}}$ |

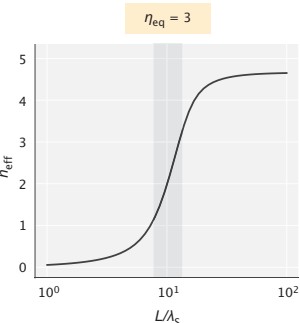 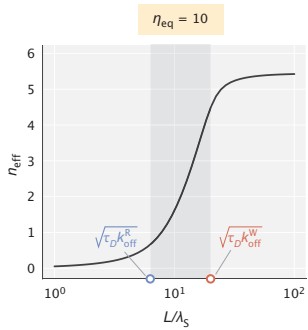 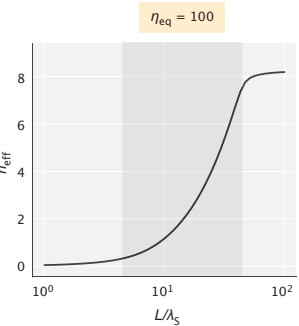

**Appendix 1—figure 1.** The effective number of proofreading realizations ($n_{\text{eff}}$) as a function of $L/\lambda_{\text{S}}$. The shaded region represents the range of $L/\lambda_{\text{S}}$ values set by the key dimensionless numbers $\sqrt{\tau_D k_{\text{off}}^{\text{R}}}$ and $\sqrt{\tau_D k_{\text{off}}^{\text{W}}}$. $\tau_D$ values chosen for the demonstration were 60, 40, and 20 (in $1/k_{\text{off}}^{\text{R}}$ units) for the three different choices of $\eta_{\text{eq}}$, respectively.

## 5. Optimal diffusion time scale for maximum fidelity

*Figure 3c* of the main text illustrated the non-monotonic dependence of fidelity on the diffusion time scale $\tau_D$ for different fixed values of $\lambda_{\text{S}}$. Here, we further explore this feature by asking what sets the optimal $\tau_D$. To gain analytical insights, we focus on the case where the system can proofread, which, as we argued in the previous section, happens when $\lambda_{\text{S}}, \lambda_{\text{ES}} < L$. Under this condition, we identified two qualitatively different regimes of complex density reduction (*Equation S26*). Namely, we found that for sufficiently fast diffusion the system acted as if the substrates were localized ideally, whereas for sufficiently slow diffusion the complex density reduction was dictated solely by $\lambda_{\text{S}}$ and did not discriminate between the two substrate kinds. These two limiting behaviors are indeed reflected in *Figure 3c* where in the low $\tau_D$ limit (fast diffusion) the family of curves matches the dotted ideal localization curve, while in the high $\tau_D$ limit (slow diffusion) all curves decay to 1, corresponding to the loss of error correction.

An intuitive approach for identifying the optimal $\tau_D$ is to slow down diffusion up to the point where the density of wrong complexes at $x = L$ approaches a plateau and effectively stops decreasing. Going past this threshold would only reduce the density of right complexes at $x = L$ and thereby, reduce the fidelity. We know from *Equation S26* that plateauing for wrong complexes happens when $\lambda_{\text{EW}} \ll \lambda_{\text{S}}$ (equivalently, $\sqrt{\tau_D k_{\text{off}}^{\text{W}}} \gg L/\lambda_{\text{S}}$). Hence, our first guess for the optimal diffusion time scale $\tau_D^*$ is

$$\tau_D^* k_{\text{off}}^{\text{W}} \sim \left(\frac{L}{\lambda_{\text{S}}}\right)^2 \Rightarrow \tag{S27}$$

$$\tau_D^* k_{\text{off}}^{\text{R}} \sim \frac{k_{\text{off}}^{\text{R}}}{k_{\text{off}}^{\text{W}}} \left(\frac{L}{\lambda_{\text{S}}}\right)^2 \Rightarrow \tag{S28}$$

$$\tau_D^* / \tau_{\text{off}}^{\text{R}} \sim \frac{1}{\eta_{\text{eq}}} \left(\frac{L}{\lambda_{\text{S}}}\right)^2. \tag{S29}$$

To test the soundness of this expression, we compared its predictions to the optimal $\tau_D$ values in *Figure 3* that were identified numerically for different choices of $\lambda_{\text{S}}$. The results of the comparison are shown in *Appendix 1—figure 2*. As can be seen, for sufficiently high degrees of substrate localization ($L/\lambda_{\text{S}}$), the prediction of *Equation S29* provides a good approximation of the true optimum. However, it is apparent that the prediction consistently underestimates the true $\tau_D^*$, which was expected since plateauing of $\rho_{\text{EW}}(L)$ happens not under equality but a strict inequality condition (i.e. $\sqrt{\tau_D^* k_{\text{off}}^{\text{W}}} \gg L/\lambda_{\text{S}}$). Because an exact analytical expression for $\tau_D^*$ is not available, we performed different approximations to the fidelity formula and found an empirical correction term for our earlier estimate given by $2(L/\lambda_{\text{S}})/\sqrt{\eta_{\text{eq}}}$. The prediction for $\tau_D^*$ with the correction term is now accurate starting

a much lower value of $L/\lambda_\mathrm{s}$, corresponding to a regime where the system proofreads once ($n_\mathrm{eff} \approx 1$). Overall, these analytical results provide good initial guesses for $\tau_D^*$ which should be refined using a numerical approach for a higher accuracy.

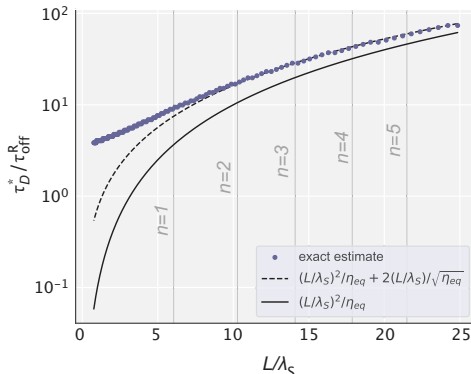

**Appendix 1—figure 2.** Optimal diffusion time scale for different choices of $\lambda_\mathrm{s}$. Blue dots represent the exact values obtained numerically for the data in **Figure 3c**. Dashed and solid lines represent the analytical estimates with and without the correction term. Vertical lines correspond to those values of $L/\lambda_\mathrm{s}$ that yield an integer number of effective proofreading realizations.

## Appendix 2

### Energetics of the scheme

We start this section by deriving an analytical expression for the minimum dissipated power, which was used in making *Figure 4* of the main text. Then, we calculate the upper limit on fidelity enhancement available to our model for a finite substrate gradient length scale and compare this limit with the fundamental thermodynamic bound. We end the section by providing an estimate for the baseline cost of setting up gradients and compare this cost with the maintenance cost reported in the main text. Similar to our treatment of Appendix 1, here too our calculations are based on the $\rho_{\mathrm{E}} \approx \mathrm{constant}$ assumption to allow for intuitive analytical results.

### 1. Derivation of the minimum dissipated power

As stated in the main text, we calculate the minimum rate of energy dissipation necessary for maintaining the substrate profiles as

$$P = \sum_{\mathrm{S=R,W}} \int_0^L j_{\mathrm{s}}(x)\mu(x)\mathrm{d}x, \qquad (S30)$$

where $j_{\mathrm{s}}(x) = k_{\mathrm{on}}\rho_{\mathrm{s}}(x)\rho_{\mathrm{E}} - k_{\mathrm{off}}^{\mathrm{S}}\rho_{\mathrm{ES}}(x)$ is the net local substrate binding flux and $\mu(x) = \mu(0) + k_{\mathrm{B}}T\ln\rho_{\mathrm{s}}(x)/\rho_{\mathrm{s}}(0) = \mu(0) - k_{\mathrm{B}}T \cdot x/\lambda_{\mathrm{s}}$ is the local chemical potential.

Our choice for the expression of power at steady state is motivated by that fact that the enzyme transport is passive and therefore, energy needs to be spent only on counteracting the local binding/unbinding events that tend to homogenize the substrate profile. To demonstrate the validity of our proposed expression more formally, we invoke the standard approaches for calculating power (*Hill, 1977*; *Zhang et al., 2012*). In particular, for a system that is described through discrete states with transition rates $k_{i \to j}$ between them, the rate of energy dissipation at steady state is given by

$$P = k_{\mathrm{B}}T \sum_{i>j}(J_{i \to j} - J_{j \to i})\ln\frac{k_{i \to j}}{k_{j \to i}}, \qquad (S31)$$

where $J_{i \to j}$ is the flux from state $i$ into state $j$. We note here that a similar expression for the rate of total entropy production involves a $\ln(J_{i \to j}/J_{j \to i})$ term (statistical forces) instead of the $\ln(k_{i \to j}/k_{j \to i})$ term (deterministic driving forces). At steady state, however, these two expressions are mathematically equivalent (*Zhang et al., 2012*). Our choice for *Equation S31* stems from the better physical intuition that it provides in our context.

So far, the description of our system has been in terms of continuous density functions. To apply *Equation S31* for calculating power, we consider the discrete-state representation of enzyme dynamics shown in *Appendix 2—figure 1*. There, space is discretized into intervals of size $\delta x$ and diffusion is represented through jumps between neighboring sites with a rate $D/\delta x^2$. What keeps the system out of equilibrium is the spatially varying substrate profile $\rho_{\mathrm{s}}(x)$.

Because forward and backward diffusive transitions have identical rates, according to *Equation S31* they will not contribute to energy dissipation (since $\ln(1) = 0$). The contribution from the remaining substrate binding/unbinding events can then be written as

$$P = k_{\mathrm{B}}T \sum_{\mathrm{S=R,W}} \sum_i \left(k_{\mathrm{on}}\rho_{\mathrm{s}}(x_i) \times \delta n_i^{\mathrm{E}} - k_{\mathrm{off}}^{\mathrm{S}} \times \delta n_i^{\mathrm{ES}}\right)\ln\frac{k_{\mathrm{on}}\rho_{\mathrm{s}}(x_i)}{k_{\mathrm{off}}^{\mathrm{S}}}, \qquad (S32)$$

where $\delta n_i^{\mathrm{E}} = \rho_{\mathrm{E}}\delta x$ and $\delta n_i^{\mathrm{ES}} = \rho_{\mathrm{ES}}(x_i)\delta x$ are the numbers of free and substrate–bound enzymes, respectively, in the $[x_i, x_i + \delta x]$ interval. In the limit of a large number of discrete spatial intervals, the sum over $i$ in *Equation S32* can be rewritten as an integral over the coordinate $x$, namely,

$$P = k_{\mathrm{B}}T \sum_{\mathrm{S=R,W}} \int_{x=0}^{\infty} \underbrace{\left(k_{\mathrm{on}}\rho_{\mathrm{s}}(x)\rho_{\mathrm{E}} - k_{\mathrm{off}}^{\mathrm{S}}\rho_{\mathrm{ES}}(x)\right)}_{j_{\mathrm{s}}(x)}\ln\frac{k_{\mathrm{on}}\rho_{\mathrm{s}}(x)}{k_{\mathrm{off}}^{\mathrm{S}}}\,\mathrm{d}x. \qquad (S33)$$

Comparing the form of *Equation S33* to that of *Equation S30* (with $\mu(x)$ substituted), one can

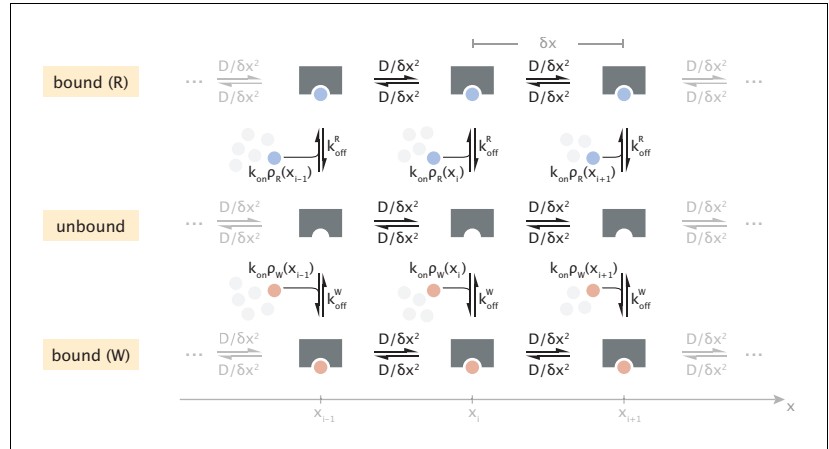

**Appendix 2—figure 1.** Discrete-state representation of diffusive transport and substrate binding/unbinding events. Transparent clusters of different numbers of substrates illustrate the spatial variation of substrate concentration.

notice a difference in the terms that multiply $j_\mathrm{s}(x)$. Specifically, in **Equation S30** we have $\mu(x) = \mu(0) - k_\mathrm{B}T \ln \rho_\mathrm{S}(0) + k_\mathrm{B}T \ln \rho_\mathrm{S}(x)$ while the corresponding term in **Equation S33** is $k_\mathrm{B}T \ln(k_\mathrm{on}/k_\mathrm{off}^\mathrm{S}) + k_\mathrm{B}T \ln \rho_\mathrm{S}(x)$. The difference between them, however, is in the parts that do not depend on $x$, while the spatially varying parts (namely, the $k_\mathrm{B}T \ln \rho_\mathrm{S}(x)$ contributions) are identical. Now, since the number of bound complexes is constant at steady state, we have $\int_0^\infty j_\mathrm{s}(x)\,\mathrm{d}x = 0$. This means that the $x$-independent parts discussed earlier all integrate to zero, making the power estimates by **Equation S30** and **Equation S33** identical, thereby justifying our proposed expression.

To estimate power, we substitute the analytical expression for $\rho_\mathrm{ES}(x)$ found earlier (**Equation S13**) into $j_\mathrm{s}(x)$ and performing a somewhat cumbersome integral, obtain

$$\beta P = J_\mathrm{bind} \sum_{\mathrm{S=R,W}} \frac{1}{1 - \lambda_\mathrm{s}^2/\lambda_\mathrm{ES}^2} \left( \frac{\lambda_\mathrm{ES}}{\lambda_\mathrm{s}} \frac{\tanh(L/2\lambda_\mathrm{ES})}{\tanh(L/2\lambda_\mathrm{s})} - 1 \right), \tag{S34}$$

where $\beta^{-1} = k_\mathrm{B}T$, and $J_\mathrm{bind} = k_\mathrm{on} S_\mathrm{total} \rho_\mathrm{E}$ is the net binding rate of each substrate. **Figure 4** in the main text was made using this expression for power.

To get additional insights about this result, let us consider the case where substrates are highly localized ($\lambda_\mathrm{s} \ll L$) and diffusion is slow ($\lambda_\mathrm{ES} \ll L$) – conditions needed for effective proofreading. Under these conditions, the hyperbolic tangent terms become 1 and the expression for the power expenditure simplifies into

$$\beta P = J_\mathrm{bind} \sum_{\mathrm{S=R,W}} \frac{\lambda_\mathrm{ES}^2}{\lambda_\mathrm{s}(\lambda_\mathrm{ES} + \lambda_\mathrm{s})}. \tag{S35}$$

The monotonic increase of power with $\lambda_\mathrm{ES}$ suggests that energy is primarily spent on maintaining the concentration gradient of right substrates. This is not surprising, since typically right complexes travel a much greater distance into the low concentration region of the compartment before releasing the bound substrate (i.e. $\lambda_\mathrm{ER} \gg \lambda_\mathrm{EW}$). Therefore, neglecting the contribution from wrong substrates and considering the range of $\lambda_\mathrm{s}$ values where the bulk of power–fidelity trade-off takes place ($\lambda_\mathrm{ER} > \lambda_\mathrm{s} > \lambda_\mathrm{EW}$), we further simplify the power expression into

$$\beta P \approx \frac{J_\mathrm{bind} \lambda_\mathrm{ER}}{\lambda_\mathrm{s}} = \frac{J_\mathrm{bind} \cdot \beta \Delta \mu}{\sqrt{\tau_D k_\mathrm{off}^\mathrm{R}}}, \tag{S36}$$

where we used the identities $\beta \Delta \mu = L/\lambda_\mathrm{s}$ and $\lambda_\mathrm{ER} = L/\sqrt{\tau_D k_\mathrm{off}^\mathrm{R}}$. This simple linear relation suggests that in order to maintain the exponential substrate profile, the minimum energy spent per substrate binding event should be at least $P/J_\mathrm{bind} \approx k_\mathrm{B}T \cdot \lambda_\mathrm{ER}/\lambda_\mathrm{s} > 1\, k_\mathrm{B}T$ (since $\lambda_\mathrm{ER} > \lambda_\mathrm{s}$).

We can also use *Equation S36* to estimate the minimum dissipation per substrate binding event at $\lambda_{\mathrm{S}} \approx \lambda_{\mathrm{EW}}$ where the logarithmic power–fidelity scaling regime ends (see *Figure 4* of the main text). Substituting the value of $\lambda_{\mathrm{S}}$, we obtain $\beta P/J_{\mathrm{bind}} \approx (\lambda_{\mathrm{ER}}/\lambda_{\mathrm{EW}}) = \sqrt{\eta_{\mathrm{eq}}}$, which is the result illustrated in *Figure 4*.

## 2. Limits on fidelity enhancement

The error reduction capacity of the spatial proofreading scheme improves with a greater difference in substrate off-rates, as was demonstrated in *Figure 2* of the main text. At the same time, *Figure 3c* showed that the finite length scale of substrate localization (or, finite driving force) sets an upper limit on fidelity enhancement for substrates with fixed off-rates. It is therefore of interest to consider these two features together to find the absolute limit on fidelity enhancement available to our model and then compare it with the fundamental bound set by thermodynamics.

Intuitively, fidelity will be enhanced the most if the density of right complexes does not decay across the compartment, while that of wrong complexes decays maximally. The first condition can be met if diffusion is fast or if the unbinding rate of right substrates is low, in which case we have

$$\rho_{\mathrm{ER}}(L) \approx \rho_{\mathrm{ER}}^{\infty},\tag{S37}$$

where $\rho_{\mathrm{ER}}^{\infty}$ is the equilibrium density of right complexes. Conversely, when the unbinding rate of wrong substrates is very large, the density of wrong complexes is maximally reduced at the right-most boundary and can be obtained from *Equation S24* by taking the $\lambda_{\mathrm{ES}} \to 0$ limit, namely,

$$
\begin{aligned}
\rho_{\mathrm{EW}}(L) &\approx \frac{k_{\mathrm{on}}\rho_{\mathrm{E}}\rho_{\mathrm{S}}(0)e^{-L/\lambda_{\mathrm{S}}}}{k_{\mathrm{off}}^{\mathrm{W}}} = \frac{k_{\mathrm{on}}\rho_{\mathrm{E}}S_{\mathrm{total}}e^{-L/\lambda_{\mathrm{S}}}}{\lambda_{\mathrm{S}}\left(1-e^{-L/\lambda_{\mathrm{S}}}\right)k_{\mathrm{off}}^{\mathrm{W}}} \\
&= \frac{k_{\mathrm{on}}\rho_{\mathrm{E}}S_{\mathrm{total}}}{k_{\mathrm{off}}^{\mathrm{W}}L} \times \frac{Le^{-L/\lambda_{\mathrm{S}}}}{\lambda_{\mathrm{S}}\left(1-e^{-L/\lambda_{\mathrm{S}}}\right)} \\
&= \rho_{\mathrm{EW}}^{\infty} \times \frac{\beta\Delta\mu\, e^{-\beta\Delta\mu}}{1-e^{-\beta\Delta\mu}}.
\end{aligned}
\tag{S38}
$$

Here, $\rho_{\mathrm{EW}}^{\infty}$ is the equilibrium density of wrong complexes, and $\beta\Delta\mu = L/\lambda_{\mathrm{S}}$ is the effective driving force of the scheme. Taking the ratio of *Equations S37 and S38*. Limits on fidelity enhancement, we obtain the largest fidelity enhancement of the scheme for the given driving force, namely,

$$\eta^{\max} = \frac{\rho_{\mathrm{ER}}(L)}{\rho_{\mathrm{EW}}(L)} = \underbrace{\frac{\rho_{\mathrm{ER}}^{\infty}}{\rho_{\mathrm{EW}}^{\infty}}}_{\eta_{\mathrm{eq}}} \times \frac{e^{\beta\Delta\mu}-1}{\beta\Delta\mu} \Rightarrow \tag{S39}$$

$$\left(\eta/\eta_{\mathrm{eq}}\right)^{\max} = (e^{\beta\Delta\mu}-1)/\beta\Delta\mu.\tag{S40}$$

When $\beta\Delta\mu \gtrsim 1$ (or, $\lambda_{\mathrm{S}} \lesssim L$), the limit above gets further simplified into

$$\left(\eta/\eta_{\mathrm{eq}}\right)^{\max} \approx e^{\beta\Delta\mu}/\beta\Delta\mu.\tag{S41}$$

Now, thermodynamics imposes an upper bound on fidelity enhancement by any proofreading scheme operating with a finite chemical potential $\Delta\mu$. This bound is equal to $e^{\beta\Delta\mu}$ and is reached when the entire chemical potential is used to increase the free energy difference between right and wrong substrates (*Qian, 2006*). Comparing it with the result in *Equation S41*, we can see that fidelity enhancement in the spatial proofreading model has the same exponential scaling term, but with an additional linear factor. Since the dominant contribution comes from the exponential term (as captured also in *Appendix 2—figure 2*), we can claim that our proposed model can operate very close to the fundamental thermodynamic limit.

## 3. Energetic cost to setup a concentration gradient

Earlier in the section, we calculated the rate at which energy needs to be dissipated to counteract the homogenizing effect that enzyme activity has on the substrate gradient. In addition to this cost,

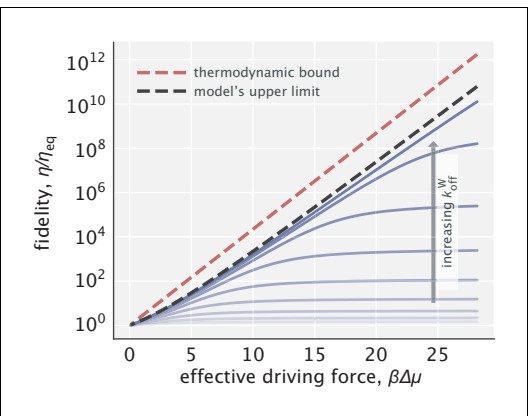

**Appendix 2—figure 2.** Fidelity enhancement as a function of the effective driving force for varying choices of $k_{off}^W$. The red dashed line indicates the thermodynamic bound given by $e^{\beta \Delta \mu}$. The black dashed line corresponds to the model's upper limit on fidelity enhancement given by **Equation S40**.

however, there is also a baseline cost for setting up a gradient in the absence of any enzyme. Here, we calculate this cost in the case where the gradient formation mechanism needs to work against diffusion that tends to flatten the substrate profile.

As before, we consider an exponentially decaying substrate gradient with a decay length scale $\lambda_s$ and a total number of substrates $S_{total}$. We write the minimum power $P_D$ required for counteracting the diffusion of substrates as

$$P_D = -\int_0^L J_D(x)\mu'(x)\,\mathrm{d}x, \tag{S42}$$

where $J_D = -D_s \nabla \rho_s(x)$ is the diffusive flux, with $D_s$ being the substrate diffusion constant. The rationale for writing this form is that diffusion moves substrates from a higher chemical potential region into a neighboring lower chemical potential region. The gradient maintaining mechanism would need to spend at least this chemical potential difference ($\delta\mu = -\mu'(x)\delta x$) per each substrate diffusing a distance $\delta x$ down the chemical potential gradient. Adding up the contribution from all local neighborhoods with a local diffusive flux $J_D(x)$ results in **Equation S42**.

Now, substituting $\rho_s(x) \sim e^{-x/\lambda_s}$ for the substrate profile and $\mu(x) = \mu(0) + k_B T \ln(\rho_s(x)/\rho_s(0))$ for the chemical potential, we obtain

$$\begin{aligned}
\beta P_D &= \int_0^L D_s \rho_s'(x)(\ln \rho_s(x))'\,\mathrm{d}x \\
&= D_s \int_0^L \frac{\left(\rho_s'(x)\right)^2}{\rho_s(x)}\,\mathrm{d}x \\
&= D_s \int_0^L \frac{\rho_s(x)}{\lambda_s^2}\,\mathrm{d}x \\
&= \frac{D_s S_{total}}{\lambda_s^2},
\end{aligned} \tag{S43}$$

where in the third step we used the relation $\rho_s'(x) = -\rho_s(x)/\lambda_s$. This suggests that the minimum dissipated power required for setting up an exponential gradient increases quadratically with decreasing localization length scale $\lambda_s$.

It is informative to also make a comparison between this result and the earlier calculated minimum dissipation needed to counteract the enzyme's homogenizing activity. Recall that when substrates were sufficiently localized and when diffusion was sufficiently slow, proofreading power could be approximated as (**Equation S35**)

$$\beta P \approx J_{\text{bind}} \frac{\lambda_{\text{ES}}^2}{\lambda_{\text{S}}(\lambda_{\text{ES}} + \lambda_{\text{S}})}, \tag{S44}$$

where $J_{\text{bind}} = k_{\text{on}} S_{\text{total}} \rho_{\text{E}}$ is the total substrate binding flux. Using the identities $\lambda_{\text{ES}} = \sqrt{D/k_{\text{off}}^{\text{S}}}$ and $K_{\text{d}}^{\text{S}} = k_{\text{off}}^{\text{S}}/k_{\text{on}}$, we can calculate the ratio of the proofreading power to baseline power as

$$\begin{aligned}
\frac{P}{P_D} &= \frac{k_{\text{on}} S_{\text{total}} \rho_{\text{E}} \lambda_{\text{ES}}^2}{D_{\text{S}} S_{\text{total}}} \times \frac{\lambda_{\text{S}}^2}{\lambda_{\text{S}}(\lambda_{\text{ES}} + \lambda_{\text{S}})} \\
&= \frac{D}{D_{\text{S}}} \times \frac{\rho_{\text{E}}}{K_{\text{d}}^{\text{S}}} \times \frac{\lambda_{\text{S}}/\lambda_{\text{ES}}}{1 + \lambda_{\text{S}}/\lambda_{\text{ES}}}.
\end{aligned} \tag{S45}$$

Presuming for simplicity that the enzyme and substrate diffusion constants are the same, we see that two factors determine the power ratio: (1) the amount of free enzyme in the system ($\rho_{\text{E}}/K_{\text{d}}^{\text{S}}$) and (2) the substrate localization length scale relative to the characteristic length scale of complex diffusion ($\lambda_{\text{S}}/\lambda_{\text{ES}}$). Now, recall that the proofreading cost is spent largely on counteracting the homogenizing activity of the enzyme on right substrates (Appendix 2.1) and that the bulk of fidelity enhancement takes place when $\lambda_{\text{S}} \lesssim \lambda_{\text{ER}}$ (Appendix 1.4). Therefore, when tuning $\lambda_{\text{S}}$ down, initially the power ratio would only depend on the amount of free enzyme in the system ($\rho_{\text{E}}/K_{\text{d}}^{\text{S}}$) and then, with tighter substrate localization, the relative contribution of the proofreading power would start to decrease.

In the end, we would like to note that spatial gradients can also be set up using an external potential without a continuous dissipation of energy. In an in vivo setting, gravity can give rise to spatial structures in oocytes (*Feric and Brangwynne, 2013*), while in an in vitro setting, electric fields can create gradients and power the transport of the complex (*Hansen et al., 2017*). We leave the investigations of such alternative strategies to future work.

## Appendix 3

### Studies on the effect of catalysis on the model performance

In Appendix 1, we considered the rate of catalysis at the right boundary to be very small for the analytical simplicity of our derivations. This resulted in expressions for fidelity that were independent of the rate of catalysis $r$ and allowed us to use the complex density at the right boundary as a proxy for speed. In this section, we relax this assumption and explore the consequences of having non-negligible catalysis rates on the model's fidelity and on the speed–fidelity trade-off.

### 1. Derivation of the complex density profile $\rho_{\mathrm{ES}}(x)$

Accounting for catalysis in our model should be done through a boundary condition for the complex density equation (*Equation S1*). Earlier, we imposed a no-flux boundary condition at $x = L$ under the slow catalysis assumption. With non-negligible catalysis, this assumption is no longer valid, and the boundary condition is modified into

$$-D\frac{\mathrm{d}\rho_{\mathrm{ES}}}{\mathrm{d}x}\Big|_{x=L} = \underbrace{r\rho_{\mathrm{ES}}(L)}_{\text{catalysis flux}} . \tag{S46}$$

Recall from *Equations S4, S6 and S8* that the general solution for the complex profile had the form

$$\rho_{\mathrm{ES}}(x) = C_1 e^{-x/\lambda_{\mathrm{ES}}} + C_2 e^{x/\lambda_{\mathrm{ES}}} + C_p e^{-x/\lambda_{\mathrm{S}}}, \quad \text{where} \tag{S47}$$

$$C_p = \frac{k_{\mathrm{on}}\rho_{\mathrm{S}}(0)\rho_{\mathrm{E}}}{k_{\mathrm{off}}^{\mathrm{S}}\left(1 - \frac{\lambda_{\mathrm{ES}}^2}{\lambda_{\mathrm{S}}^2}\right)} . \tag{S48}$$

Imposing the no-flux boundary condition at $x = 0$ allows us to eliminate one of the integration constants, namely,

$$-D\frac{\mathrm{d}\rho_{\mathrm{ES}}}{\mathrm{d}x}\Big|_{x=0} = -D\left(-\frac{C_1}{\lambda_{\mathrm{ES}}} + \frac{C_2}{\lambda_{\mathrm{ES}}} - \frac{C_p}{\lambda_{\mathrm{S}}}\right) = 0 \Rightarrow \tag{S49}$$

$$C_2 = C_1 + \frac{\lambda_{\mathrm{ES}}}{\lambda_{\mathrm{S}}}C_p \Rightarrow \tag{S50}$$

$$\rho_{\mathrm{ES}}(x) = C_1\left(e^{-x/\lambda_{\mathrm{ES}}} + e^{x/\lambda_{\mathrm{ES}}}\right) + \frac{C_p}{\lambda_{\mathrm{S}}}\left(\lambda_{\mathrm{ES}}e^{x/\lambda_{\mathrm{ES}}} + \lambda_{\mathrm{S}}e^{-x/\lambda_{\mathrm{S}}}\right)$$

$$= 2C_1\cosh(x/\lambda_{\mathrm{ES}}) + \frac{C_p}{\lambda_{\mathrm{S}}}\left(\lambda_{\mathrm{ES}}e^{x/\lambda_{\mathrm{ES}}} + \lambda_{\mathrm{S}}e^{-x/\lambda_{\mathrm{S}}}\right). \tag{S51}$$

Next, we impose the new boundary condition at $x = L$ (*Equation S46*), which yields

$$-D\left(\frac{2C_1}{\lambda_{\mathrm{ES}}}\sinh(L/\lambda_{\mathrm{ES}}) + \frac{C_p}{\lambda_{\mathrm{S}}}\left(e^{L/\lambda_{\mathrm{ES}}} - e^{-L/\lambda_{\mathrm{S}}}\right)\right)$$

$$= r\left(2C_1\cosh(L/\lambda_{\mathrm{ES}}) + \frac{C_p}{\lambda_{\mathrm{S}}}\left(\lambda_{\mathrm{ES}}e^{L/\lambda_{\mathrm{ES}}} + \lambda_{\mathrm{S}}e^{-L/\lambda_{\mathrm{S}}}\right)\right)$$

$$\Rightarrow 2C_1\sinh(L/\lambda_{\mathrm{ES}}) + \frac{C_p}{\lambda_{\mathrm{S}}}\lambda_{\mathrm{ES}}\left(e^{L/\lambda_{\mathrm{ES}}} - e^{-L/\lambda_{\mathrm{S}}}\right) \tag{S52}$$

$$= -\underbrace{\frac{\lambda_{\mathrm{ES}}r}{D}}_{\varepsilon}\left(2C_1\cosh(L/\lambda_{\mathrm{ES}}) + \frac{C_p}{\lambda_{\mathrm{S}}}\left(\lambda_{\mathrm{ES}}e^{L/\lambda_{\mathrm{ES}}} + \lambda_{\mathrm{S}}e^{-L/\lambda_{\mathrm{S}}}\right)\right).$$

Note that we have introduced the dimensionless variable $\varepsilon$, which, as will see later, will define the extent to which the presence of catalysis affects the fidelity. For convenience, here we write different equivalent forms for $\varepsilon$ as

$$\varepsilon = \frac{\lambda_{\mathrm{ES}}r}{D} = \frac{r}{\sqrt{Dk_{\mathrm{off}}^{\mathrm{S}}}} = \frac{r}{Lk_{\mathrm{off}}^{\mathrm{S}}}\sqrt{\tau_D k_{\mathrm{off}}^{\mathrm{S}}}. \tag{S53}$$

Solving for the remaining unknown coefficient $C_1$ in *Equation S52*, we find

$$C_1 = -\frac{C_p}{2\lambda_{\mathrm{S}}} \frac{\lambda_{\mathrm{ES}}\left(e^{L/\lambda_{\mathrm{ES}}} - e^{-L/\lambda_{\mathrm{S}}}\right) + \varepsilon\left(\lambda_{\mathrm{ES}} e^{L/\lambda_{\mathrm{ES}}} + \lambda_{\mathrm{S}} e^{-L/\lambda_{\mathrm{S}}}\right)}{\sinh(L/\lambda_{\mathrm{ES}}) + \varepsilon\cosh(L/\lambda_{\mathrm{ES}})}. \tag{S54}$$

Lastly, we substitute this result for $C_1$ into **Equation S51** and obtain a general expression for the complex density profile as

$$\rho_{\mathrm{ES}}(x) = -\frac{C_p}{\lambda_{\mathrm{S}}} \frac{\lambda_{\mathrm{ES}}\left(e^{L/\lambda_{\mathrm{ES}}} - e^{-L/\lambda_{\mathrm{S}}}\right) + \varepsilon\left(\lambda_{\mathrm{ES}} e^{L/\lambda_{\mathrm{ES}}} + \lambda_{\mathrm{S}} e^{-L/\lambda_{\mathrm{S}}}\right)}{\sinh(L/\lambda_{\mathrm{ES}}) + \varepsilon\cosh(L/\lambda_{\mathrm{ES}})} \cosh(x/\lambda_{\mathrm{ES}}) + \frac{C_p}{\lambda_{\mathrm{S}}}\left(\lambda_{\mathrm{ES}} e^{x/\lambda_{\mathrm{ES}}} + \lambda_{\mathrm{S}} e^{-x/\lambda_{\mathrm{S}}}\right). \tag{S55}$$

One can show in a straightforward way that this result reduces to **Equation S13** in the $\varepsilon \to 0$ limit.

## 2. Effects on fidelity in low and high substrate localization regimes

Accounting for the catalysis flux has made the general expression for the complex density profile even more incomprehensible. In order to gain insights about the qualitative as well as quantitative changes introduced by catalysis, we will focus on two characteristic limits of substrate localization – uniform substrate profile ($\lambda_{\mathrm{S}} \to \infty$) and ideal substrate localization ($\lambda_{\mathrm{S}} \to 0$).

### 2.1. Uniform substrate profile

In this case, no mechanism for localizing substrates is in play. Let us start off by evaluating the coefficient $C_p$ (**Equation S48**) in the $\lambda_{\mathrm{S}} \to \infty$ limit. Recalling from **Equation S3** that $\rho_{\mathrm{S}}(0) = S_{\mathrm{total}}/(\lambda_{\mathrm{S}}(1 - e^{-L/\lambda_{\mathrm{S}}}))$, we find

$$\rho_{\mathrm{S}}(0) \approx \frac{S_{\mathrm{total}}}{L} \Rightarrow \tag{S56}$$

$$C_p \approx \frac{k_{\mathrm{on}}\rho_{\mathrm{S}}(0)\rho_{\mathrm{E}}}{k_{\mathrm{off}}^{\mathrm{S}}}$$

$$\approx \frac{k_{\mathrm{on}}S_{\mathrm{total}}\rho_{\mathrm{E}}}{L k_{\mathrm{off}}^{\mathrm{S}}}$$

$$= \frac{J_{\mathrm{bind}}}{L k_{\mathrm{off}}^{\mathrm{S}}}, \tag{S57}$$

where $J_{\mathrm{bind}} = k_{\mathrm{on}}S_{\mathrm{total}}\rho_{\mathrm{E}}$ is the total substrate binding flux.

Substituting the expression for $C_p$ into **Equation S55** and eliminating all the terms that vanish upon taking the $\lambda_{\mathrm{S}} \to \infty$ limit, we obtain

$$\rho_{\mathrm{ES}}(x) \approx C_p\left(1 - \frac{\varepsilon\cosh(x/\lambda_{\mathrm{ES}})}{\sinh(L/\lambda_{\mathrm{ES}}) + \varepsilon\cosh(L/\lambda_{\mathrm{ES}})}\right)$$

$$= \frac{J_{\mathrm{bind}}}{L k_{\mathrm{off}}^{\mathrm{S}}} \times \frac{\sinh(L/\lambda_{\mathrm{ES}}) + \varepsilon(\cosh(L/\lambda_{\mathrm{ES}}) - \cosh(x/\lambda_{\mathrm{ES}}))}{\sinh(L/\lambda_{\mathrm{ES}}) + \varepsilon\cosh(L/\lambda_{\mathrm{ES}})}. \tag{S58}$$

Ultimately, we are interested in knowing the rate of product formation defined via $v_{\mathrm{S}} = r\rho_{\mathrm{ES}}(L)$. We therefore evaluate the complex density at $x = L$ and multiply it by $r$, which yields

$$v_{\mathrm{S}} = r\rho_{\mathrm{ES}}(L) = J_{\mathrm{bind}} \times \left(\frac{r}{L k_{\mathrm{off}}^{\mathrm{S}}}\right) \times \frac{\sinh(L/\lambda_{\mathrm{ES}})}{\sinh(L/\lambda_{\mathrm{ES}}) + \varepsilon\cosh(L/\lambda_{\mathrm{ES}})}$$

$$= J_{\mathrm{bind}} \times \left(\frac{r}{L k_{\mathrm{off}}^{\mathrm{S}}}\right) \times \frac{\tanh(L/\lambda_{\mathrm{ES}})}{\tanh(L/\lambda_{\mathrm{ES}}) + \varepsilon} \tag{S59}$$

$$\equiv J_{\mathrm{bind}} \times \left(\frac{r}{L k_{\mathrm{off}}^{\mathrm{S}}}\right) \times \frac{\tanh\left(\sqrt{\tau_D k_{\mathrm{off}}^{\mathrm{S}}}\right)}{\tanh\left(\sqrt{\tau_D k_{\mathrm{off}}^{\mathrm{S}}}\right) + \varepsilon},$$

where in the last step we wrote an equivalent expression using the $L/\lambda_{\mathrm{ES}} = \sqrt{\tau_D k_{\mathrm{off}}^{\mathrm{S}}}$ identity. To analyze this result further, we will consider two limiting cases.

**Case 1: Fast diffusion** ($\sqrt{\tau_D k_{\text{off}}^{\text{S}}} \ll 1$). If diffusion is fast, we can approximate the hyperbolic tangent functions as the arguments themselves (i.e. $\tanh(z) \approx z$ for $z \ll 1$). Then, using the last form of $\varepsilon$ in *Equation S53*, we simplify the expression for speed as

$$v_{\text{S}} \approx J_{\text{bind}} \times \left(\frac{r}{L k_{\text{off}}^{\text{S}}}\right) \times \frac{\sqrt{\tau_D k_{\text{off}}^{\text{S}}}}{\sqrt{\tau_D k_{\text{off}}^{\text{S}}} + \frac{r}{L k_{\text{off}}^{\text{S}}}\sqrt{\tau_D k_{\text{off}}^{\text{S}}}}$$

$$= J_{\text{bind}} \times \left(\frac{r}{L k_{\text{off}}^{\text{S}}}\right) \times \frac{1}{1 + \frac{r}{L k_{\text{off}}^{\text{S}}}} \Rightarrow \tag{S60}$$

$$v_{\text{S}} = J_{\text{bind}} \times \frac{\tilde{r}}{k_{\text{off}}^{\text{S}} + \tilde{r}}, \quad \text{where} \tag{S61}$$

$$\tilde{r} = r/L. \tag{S62}$$

This is an intuitive result, suggesting that an enzyme that diffuses fast acts like a standard Michaelis–Menten enzyme with an effective catalysis rate $\tilde{r}$. For such an enzyme, the probability of catalysis for a bound substrate is $\tilde{r}/(k_{\text{off}}^{\text{S}} + \tilde{r})$. Multiplying this probability by the net substrate binding flux yields the expression for speed in *Equation S61*.

Fidelity of the model in this fast diffusion setting can be written as

$$\eta = \frac{v_{\text{R}}}{v_{\text{W}}} = \frac{k_{\text{off}}^{\text{W}} + \tilde{r}}{k_{\text{off}}^{\text{R}} + \tilde{r}}. \tag{S63}$$

In the limit where catalysis is very slow ($\tilde{r} \ll k_{\text{off}}^{\text{R}}$), the equilibrium fidelity given by the ratio of off-rates is recovered. And in the opposite limit of very fast catalysis ($\tilde{r} \gg k_{\text{off}}^{\text{W}}$), the discriminatory capacity of the enzyme disappears altogether (*Appendix 3—figure 1a*).

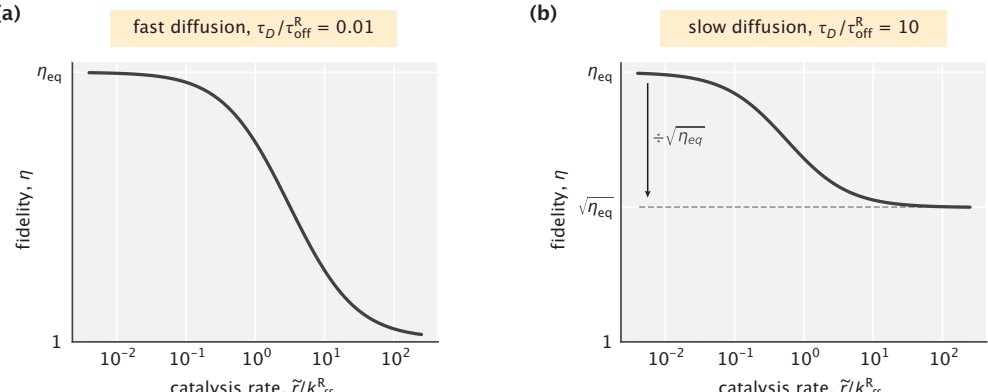

**Appendix 3—figure 1.** Dependence of fidelity on the catalysis rate in the case where the substrate profile is uniform. (**a**) Fast diffusion setting ($\sqrt{\tau_D k_{\text{off}}^{\text{R}}} \ll 1$). The highest fidelity reduction is a factor of $\eta_{\text{eq}}$. (**b**) Slow diffusion setting ($\sqrt{\tau_D k_{\text{off}}^{\text{R}}} \gtrsim 1$). The highest fidelity reduction is a factor of $\sqrt{\eta_{\text{eq}}}$. In both cases, $\eta_{eq} = 10$ was used.

**Case 2: Slow diffusion** ($\sqrt{\tau_D k_{\text{off}}^{\text{S}}} \gtrsim 1$). A more interesting case is when diffusion is slow. Now, the hyperbolic tangent functions in *Equation S59* are approximately 1, allowing us to simplify the expression for speed into

$$v_{\text{S}} = J_{\text{bind}} \times \left(\frac{r}{L k_{\text{off}}^{\text{S}}}\right) \times \frac{1}{1 + \frac{r}{L k_{\text{off}}^{\text{S}}}\sqrt{\tau_D k_{\text{off}}^{\text{S}}}}$$

$$= J_{\text{bind}} \times \frac{\tilde{r}}{k_{\text{off}}^{\text{S}} + \tilde{r}\sqrt{\tau_D k_{\text{off}}^{\text{S}}}}. \tag{S64}$$

Drawing an analogy between the above result and *Equation S61*, one can notice the presence of an extra $\sqrt{\tau_D k_{\text{off}}^{\text{S}}}$ factor for $\tilde{r}$ in the denominator.

Evaluating the speeds of right and wrong product formation, we can write fidelity in this slow diffusion setting as

$$\eta = \frac{v_{\text{R}}}{v_{\text{W}}} = \frac{k_{\text{off}}^{\text{W}} + \tilde{r}\sqrt{\tau_D k_{\text{off}}^{\text{W}}}}{k_{\text{off}}^{\text{R}} + \tilde{r}\sqrt{\tau_D k_{\text{off}}^{\text{R}}}}. \tag{S65}$$

Like the fast diffusion case, when catalysis is very slow ($\tilde{r} \ll \sqrt{k_{\text{off}}^{\text{R}}/\tau_D}$ or, equivalently, $r \ll \sqrt{D k_{\text{off}}^{\text{R}}}$), the equilibrium fidelity is recovered. Unlike the fast diffusion case, however, if catalysis is very fast ($r \gg \sqrt{D k_{\text{off}}^{\text{W}}}$), the enzyme partly preserves its discriminatory capacity (*Appendix 3—figure 1b*). In this limit, a fidelity equal to the square root of the equilibrium fidelity is still attainable, namely,

$$\eta \approx \frac{\sqrt{k_{\text{off}}^{\text{W}}}}{\sqrt{k_{\text{off}}^{\text{R}}}} = \sqrt{\eta_{\text{eq}}}. \tag{S66}$$

This unexpected result suggests a potential advantage of localizing fast catalytic reactions instead of having them occur in a well–mixed solution.

## 2.2. Ideal substrate localization

We next consider the effect of catalysis on model fidelity in the ideal substrate localization limit ($\lambda_{\text{S}} \to 0$). We begin by evaluating the $C_p/\lambda_{\text{S}}$ ratio that appears in the density profile expression (*Equation S55*). Using *Equations S48* and *Equations S3*, we find

$$\rho_{\text{S}}(0) \approx \frac{S_{\text{total}}}{\lambda_{\text{S}}} \tag{S67}$$

$$\begin{aligned}
\frac{C_p}{\lambda_{\text{S}}} &= \frac{k_{\text{on}}\rho_{\text{S}}(0)\rho_{\text{E}}}{\lambda_{\text{S}} k_{\text{off}}^{\text{S}}(1 - \lambda_{\text{ES}}^2/\lambda_{\text{S}}^2)} \\
&\approx \frac{k_{\text{on}}S_{\text{total}}\rho_{\text{E}}}{-k_{\text{off}}^{\text{S}}\lambda_{\text{ES}}^2} \\
&= -\frac{J_{\text{bind}}}{D},
\end{aligned} \tag{S68}$$

where in the last step we invoked the identities $\lambda_{\text{ES}}^2 = D/k_{\text{off}}^{\text{S}}$ and $J_{\text{bind}} = k_{\text{on}}S_{\text{total}}\rho_{\text{E}}$. We then substitute our result for $C_p/\lambda_{\text{S}}$ into *Equation S55* and simplify the complex density expression into

$$\begin{aligned}
\rho_{\text{ES}}(x) &= \frac{J_{\text{bind}}}{D} \times \lambda_{\text{ES}} \left( \frac{e^{L/\lambda_{\text{ES}}} + \varepsilon e^{L/\lambda_{\text{ES}}}}{\sinh(L/\lambda_{\text{ES}}) + \varepsilon \cosh(L/\lambda_{\text{ES}})}\cosh(x/\lambda_{\text{ES}}) - e^{x/\lambda_{\text{ES}}} \right) \\
&= J_{\text{bind}} \times \frac{\lambda_{\text{ES}}}{D} \frac{\cosh((L-x)/\lambda_{\text{ES}}) + \varepsilon \sinh((L-x)/\lambda_{\text{ES}})}{\sinh(L/\lambda_{\text{ES}}) + \varepsilon \cosh(L/\lambda_{\text{ES}})}.
\end{aligned} \tag{S69}$$

To obtain the speed, we evaluate $\rho_{\text{ES}}(x)$ at the right boundary ($x = L$) and multiply it by $r$, namely,

$$\begin{aligned}
v_{\text{S}} = r\rho_{\text{ES}}(L) &= J_{\text{bind}} \underbrace{\frac{\lambda_{\text{ES}} r}{D}}_{\varepsilon} \frac{1}{\sinh(L/\lambda_{\text{ES}}) + \varepsilon \cosh(L/\lambda_{\text{ES}})} \\
&= J_{\text{bind}} \times \frac{\varepsilon}{\sinh\left(\sqrt{\tau_D k_{\text{off}}^{\text{S}}}\right) + \varepsilon \cosh\left(\sqrt{\tau_D k_{\text{off}}^{\text{S}}}\right)}.
\end{aligned} \tag{S70}$$

To evaluate the effect of catalysis further, we again consider two special limits – those of fast and slow diffusion.

**Case 1: Fast diffusion** ($\sqrt{\tau_D k_{off}^S} \ll 1$). In this limit, the hyperbolic sine function can be approximated by its argument (i.e. $\sinh(z) \approx z$ for $z \ll 1$), while the hyperbolic cosine function is approximately 1. Making these approximations and substituting the expression for $\varepsilon$, we obtain

$$v_S \approx J_{\text{bind}} \times \frac{\frac{r}{Lk_{off}^S}\sqrt{\tau_D k_{off}^S}}{\sqrt{\tau_D k_{off}^S} + \frac{r}{Lk_{off}^S}\sqrt{\tau_D k_{off}^S}}$$

$$= J_{\text{bind}} \times \frac{\frac{r}{Lk_{off}^S}}{1 + \frac{r}{Lk_{off}^S}}$$

$$= J_{\text{bind}} \times \frac{\tilde{r}}{k_{off}^S + \tilde{r}}. \tag{S71}$$

This result is identical to what we found in the fast diffusion limit for the $\lambda_s \to \infty$ setting (*Equation S61*), which is reasonable, since the location of substrate binding is irrelevant if diffusion is very fast (*Appendix 3—figure 2a*).

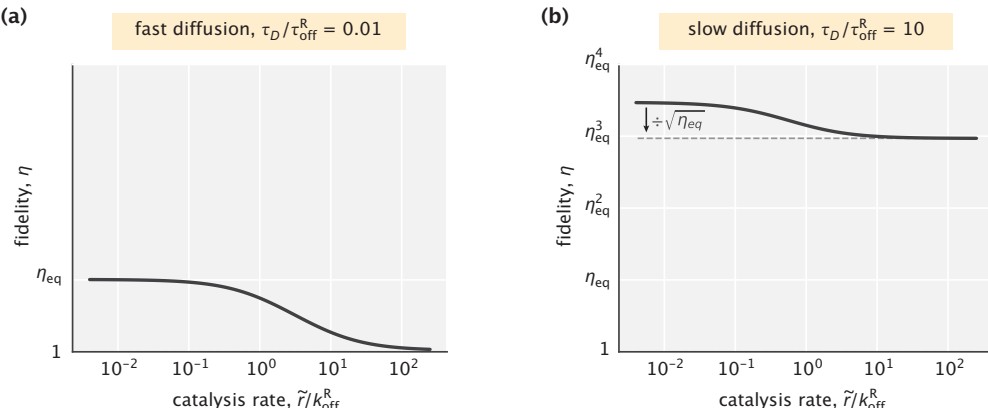

**(a)** fast diffusion, $\tau_D/\tau_{off}^R = 0.01$ **(b)** slow diffusion, $\tau_D/\tau_{off}^R = 10$

**Appendix 3—figure 2.** Fidelity as a function of the catalysis rate in an ideal substrate localization setting. (**a**) Fast diffusion case, where the behavior of the system is identical to that in *Appendix 3—figure 1a*. (**b**) Slow diffusion case where efficient proofreading is achieved. Catalysis can reduce the fidelity by up to a factor of $\sqrt{\eta_{\text{eq}}}$. In both cases, $\eta_{eq} = 10$ was used.

**Case 2: Slow diffusion** ($\sqrt{\tau_D k_{off}^S} \gg 1$). In this limit, the hyperbolic sine and cosine functions can be approximated as exponentials with a $1/2$ prefactor, simplifying the expression of speed into

$$v_S \approx J_{\text{bind}} \times \frac{2\varepsilon}{1+\varepsilon} e^{-\sqrt{\tau_D k_{off}^S}}. \tag{S72}$$

Recalling the identity $\varepsilon = r/\sqrt{Dk_{off}^S}$ (note that $\varepsilon$ depends on the substrate kind), we evaluate the speed for right and wrong product formation and, dividing them, obtain the fidelity as

$$\eta = \frac{v_R}{v_W} = \frac{1 + r/\sqrt{Dk_{off}^W}}{1 + r/\sqrt{Dk_{off}^R}} \times \frac{\sqrt{k_{off}^W}}{\sqrt{k_{off}^R}} e^{\sqrt{\tau_D k_{off}^W} - \sqrt{\tau_D k_{off}^R}}$$

$$= \frac{1 + r/\sqrt{Dk_{off}^W}}{1 + r/\sqrt{Dk_{off}^R}} \times \sqrt{\eta_{\text{eq}}} e^{\sqrt{\tau_D k_{off}^R}(\sqrt{\eta_{\text{eq}}} - 1)}. \tag{S73}$$

In the case where catalysis is slow ($r \ll \sqrt{Dk_{off}^R}$), the first term in the fidelity expression becomes approximately 1, and the our earlier result obtained with no account of catalysis is recovered (*Equation S21*). In the opposite limit of fast catalysis ($r \gg \sqrt{Dk_{off}^W}$), the first term is no longer 1, and we find

$$\eta \approx \underbrace{\sqrt{\frac{k_{\text{off}}^{\text{R}}}{k_{\text{off}}^{\text{W}}}} \sqrt{\eta_{\text{eq}}}}_{1} e^{\sqrt{\tau_D k_{\text{off}}^{\text{R}}}(\sqrt{\eta_{\text{eq}}}-1)}$$

(S74)

$$= e^{\sqrt{\tau_D k_{\text{off}}^{\text{R}}}(\sqrt{\eta_{\text{eq}}}-1)}.$$

As we can see, fast catalysis in the slow diffusion regime reduces the fidelity by $\sqrt{\eta_{\text{eq}}}$ or, equivalently, reduces the effective number of proofreading realizations by one half, without affecting the exponential amplification term (*Appendix 3—figure 2b*).

To conclude, our study demonstrated the expected reduction of fidelity with increasing catalysis rate. In the case of fast diffusion, up to a factor of $\eta_{\text{eq}}$ reduction is possible, as is the case for the original (*Hopfield, 1974*; *Wong et al., 2018*). In the case of slow diffusion, however, the cap on the amount of reduction is decreased down to $\sqrt{\eta_{\text{eq}}}$. The advantage of this feature is most notable in the limit of a non-localized (i.e. uniform) substrate profile and fast catalysis where a diffusing enzyme is still capable of discriminating between substrates. This behavior would not be possible for a Michaelis–Menten enzyme in a well-mixed solution.

## 3. Effects on the speed–fidelity trade-off

In *Figure 3a* of the main text we explored the speed–fidelity trade-off in the slow catalysis limit. This trade-off arose in response to tuning the substrate localization length scale ($\lambda_s$) and the diffusion time scale ($\tau_D$). Here, we explore the changes to this trade-off behavior in the case where the effects of catalysis are not negligible. For concreteness, we focus on alterations to the Pareto front of the trade-off achieved in the $\lambda_s \to 0$ limit.

*Appendix 3—figure 3a* compares the Pareto fronts in the cases of slow and fast catalysis limits. In each case, speed is normalized by the corresponding effective Michaelis–Menten speed that is reached in the fast diffusion limit and is given by $v_{\text{MM}} = J_{\text{bind}} \times \tilde{r}/(k_{\text{off}}^{\text{R}} + \tilde{r})$, where $\tilde{r} = r/L$. One can notice a shift of the fast catalysis front toward the low-fidelity region, which was expected since earlier we observed the complete loss of substrate discrimination when diffusion and catalysis were both fast (*Appendix 3—figure 2a*).

**(a)** 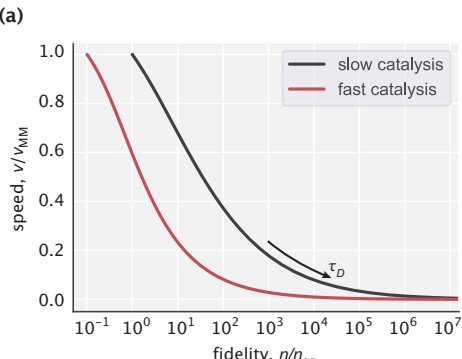   **(b)** 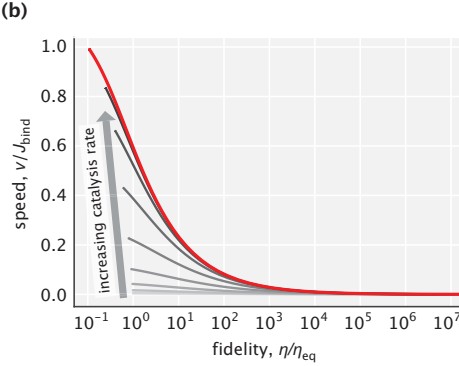

**Appendix 3—figure 3.** Pareto front of the speed–fidelity trade-off at different levels of catalytic activity. (**a**) Cases of slow and fast catalysis limits, with the *y*-axis for speed normalized to the [0,1] interval. (**b**) Family of Pareto fronts for different choices of the catalysis rate. Speed on the *y*-axis is reported relative to the substrate binding flux $J_{\text{bind}}$.

*Appendix 3—figure 3a* may leave an impression that faster catalysis leads to a less favorable speed–fidelity trade-off. Note, however, that the speed $v_{\text{MM}}(\tilde{r})$ used to normalize the *y*-axis is itself a function of the catalysis rate and penalizes the fast catalysis case more than its slow counterpart. To eliminate this ambiguity, we plotted a family of Pareto fronts for increasing values of the catalysis rate but this time normalizing the *y*-axis by the $r$-independent quantity $J_{\text{bind}}$ (*Appendix 3—figure 3b*). As can be seen, faster catalysis in fact improves the speed–fidelity trade-off, meaning that in order to maximize fidelity at a given speed level, the best strategy would be to increase the catalysis rate and correspondingly slow down the diffusion.

A trade-off between speed and fidelity also arises in response to the sole alteration of the catalysis rate, while keeping the rest of the model parameters fixed. To explore this trade-off for an arbitrary fixed choice of $\lambda_S$ and $\tau_D$, we begin by evaluating speed from *Equation S55*, namely,

$$
\begin{aligned}
v_S &= r\rho_{ES}(L) \\
&= r \times \left( -\frac{C_p\lambda_{ES}\left(e^{L/\lambda_{ES}} - e^{-L/\lambda_S}\right) + \varepsilon\left(\lambda_{ES}e^{L/\lambda_{ES}} + \lambda_S e^{-L/\lambda_S}\right)}{\sinh(L/\lambda_{ES}) + \varepsilon\cosh(L/\lambda_{ES})}\cosh(L/\lambda_{ES}) + \frac{C_p}{\lambda_S}\left(\lambda_{ES}e^{L/\lambda_{ES}} + \lambda_S e^{-L/\lambda_S}\right) \right) \\
&= r \times \frac{C_p}{\lambda_S}\left( \frac{-\lambda_{ES}\left(e^{L/\lambda_{ES}} - e^{-L/\lambda_S}\right) \times \cosh(L/\lambda_{ES}) + \left(\lambda_{ES}e^{L/\lambda_{ES}} + \lambda_S e^{-L/\lambda_S}\right) \times \sinh(L/\lambda_{ES})}{\sinh(L/\lambda_{ES}) + \varepsilon\cosh(L/\lambda_{ES})} + \right. \\
&\qquad \left. \varepsilon \times \underbrace{\frac{-\left(\lambda_{ES}e^{L/\lambda_{ES}} + \lambda_S e^{-L/\lambda_S}\right) \times \cosh(L/\lambda_{ES}) + \left(\lambda_{ES}e^{L/\lambda_{ES}} + \lambda_S e^{-L/\lambda_S}\right) \times \cosh(L/\lambda_{ES})}{\sinh(L/\lambda_{ES}) + \varepsilon\cosh(L/\lambda_{ES})}}_{=0} \right) \\
&= r \times \frac{C_p}{\lambda_S}\frac{\left(\lambda_S\sinh(L/\lambda_{ES}) + \lambda_{ES}\cosh(L/\lambda_{ES})\right)e^{-L/\lambda_S} - \lambda_{ES}}{\sinh(L/\lambda_{ES}) + \varepsilon\cosh(L/\lambda_{ES})} \\
&= \frac{a_S r}{1 + b_S r}.
\end{aligned}
\tag{S75}
$$

In the last step, we introduced coefficients $a_S$ and $b_S$ that are *independent* from $r$, and used the fact that $\varepsilon \sim r$.

Now, using the definition of fidelity and the result obtained above, we can write

$$
\eta = \frac{v_R}{v_W} = \frac{a_R}{a_W}\frac{1 + b_W r}{1 + b_R r}.
\tag{S76}
$$

Notice that the ratio $a_R/a_W \equiv \eta_0$ is the fidelity in the limit of very slow catalysis ($r \to 0$). Substituting it, we write

$$
\begin{aligned}
\eta &= \eta_0\left(\frac{1 + b_R r - (b_R - b_W)r}{1 + b_R r}\right) \\
&= \eta_0\left(1 - (b_R - b_W) \times \underbrace{\frac{r}{1 + b_R r}}_{v_R/a_R}\right) \Rightarrow
\end{aligned}
\tag{S77}
$$

$$
\eta = \eta_0\left(1 - \frac{\Delta b}{a_R}v_R\right),
\tag{S78}
$$

where $\Delta b = b_R - b_W$. Recalling that $\varepsilon = \lambda_{ES}r/D$ and noting the function form of the denominator in *Equation S75*, one can show that $b_S = D^{-1}\lambda_{ES}/\tanh(L/\lambda_{ES})$. This is an increasing function of $\lambda_{ES}$ and hence, a decreasing function of $k_{off}^S$, implying that $\Delta b > 0$.

With this condition in mind, we can see from *Equation S78* that speed and fidelity are anticorrelated with a linear slope when tuning the catalysis rate, unlike the more sophisticated trade-off relations when tuning the other model parameters. The peak fidelity $\eta_0$ is attained in the limit of vanishing speed. And conversely, speed is the highest when fidelity is the lowest for the given fixed values of $\lambda_S$ and $\tau_D$ (*Appendix 3—figure 4*).

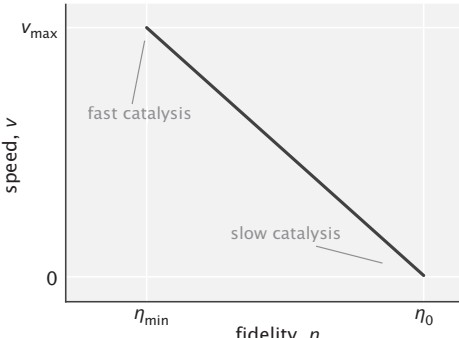

**Appendix 3—figure 4.** Linear trade-off between speed and fidelity when tuning the rate of catalysis. $\eta_{\min}$ is the fidelity in the fast catalysis limit and is up to $\eta_{eq}$ lower than $\eta_0$ (based on the results of the previous section). Linear scale is used for both axes.

Overall, our result illustrates the simple speed–fidelity trade-off that can be navigated by tuning the catalysis rate. This, for instance, can be achieved by changing the concentration of effectors that activate the enzyme for catalysis.

## Appendix 4

### Proofreading for substrates with different localization conditions

Following the original treatment by *Hopfield, 1974*, we have performed the studies of our model under the assumption that discrimination between right and wrong substrates is solely based on their off–rates ($k_{\text{off}}^{\text{W}} > k_{\text{off}}^{\text{R}}$). Although this is often the signature difference between substrates, in a cellular setting substrate discrimination may occur through other factors also. For example, substrates may be present at different amounts or they may have non-identical on–rates. These differences, however, have a multiplicative effect on the fidelity (i.e. $\eta \sim (k_{\text{on}}^{\text{R}}[\text{R}])/(k_{\text{on}}^{\text{W}}[\text{W}])$) and do not highlight the proofreading capacity of a particular model.

Unlike these two features, differences in the degree to which right and wrong substrates are localized can have a non-trivial effect on the proofreading performance. In this Appendix, we generalize our study of the model fidelity to cases where right and wrong substrates have unequal localization length scales $\lambda_{\text{R}}$ and $\lambda_{\text{W}}$, respectively.

### 1. Limiting cases

We start off by exploring the limiting cases first. From the earlier derived *Equation S14* and *Equation S15*, we know that the complex density at $x = L$ in very low ($\lambda_{\text{S}} \gg L$) and very high ($\lambda_{\text{S}} \ll L$) substrate localization regimes is given by

$$\rho_{\text{ES}}^{\infty} = \frac{k_{\text{on}} \bar{\rho}_{\text{S}} \rho_{\text{E}}}{k_{\text{off}}^{\text{S}}} \quad \text{and} \tag{S79}$$

$$\rho_{\text{ES}}^{\text{ideal}}(L) = \rho_{\text{ES}}^{\infty} \times \frac{L/\lambda_{\text{ES}}}{\sinh(L/\lambda_{\text{ES}})}, \tag{S80}$$

respectively. Note that the complex density in the ideal localization case is necessarily lower than that in the case of a uniform profile, since the inequality $\sinh(L/\lambda_{\text{ES}}) > L/\lambda_{\text{ES}}$ holds for all choices of $\lambda_{\text{ES}}$. If $\lambda_{\text{R}}$ and $\lambda_{\text{W}}$ are not constrained to be equal, then the highest fidelity for a given $\tau_D$ will be attained when the right substrates are distributed uniformly while the wrong substrates are highly localized ($\lambda_{\text{R}} \gg L$ and $\lambda_{\text{W}} \ll L$, respectively). We obtain the fidelity in this case as

$$
\begin{aligned}
\eta^{\text{max}} &= \frac{\rho_{\text{ER}}^{\infty}}{\rho_{\text{EW}}^{\text{ideal}}(L)} \\
&= \frac{\rho_{\text{ER}}^{\infty}}{\rho_{\text{EW}}^{\infty}} \times \frac{\sinh(L/\lambda_{\text{EW}})}{L/\lambda_{\text{EW}}} \\
&= \eta_{\text{eq}} \times \frac{\sinh(L/\lambda_{\text{EW}})}{L/\lambda_{\text{EW}}} \Rightarrow
\end{aligned}
\tag{S81}
$$

$$\frac{\eta^{\text{max}}}{\eta_{\text{eq}}} = \frac{\sinh(L/\lambda_{\text{EW}})}{L/\lambda_{\text{EW}}} \equiv \frac{\sinh\left(\sqrt{\tau_D k_{\text{off}}^{\text{W}}}\right)}{\sqrt{\tau_D k_{\text{off}}^{\text{W}}}}. \tag{S82}$$

Notably, this result for maximum fidelity enhancement is independent of $k_{\text{off}}^{\text{R}}$. Furthermore, it exceeds the ideal localization fidelity reported in the main text (*Equation 5*, derived in the $\lambda_{\text{S}} \to 0$ limit), which was expected since now the right complexes on average travel a shorter distance to reach the activation site than the wrong complexes.

In the opposite scenario where the wrong substrates are uniformly distributed and the right ones are highly localized ($\lambda_{\text{R}} \ll L$ and $\lambda_{\text{W}} \gg L$, respectively), the system attains its lowest fidelity for a given $\tau_D$, namely,

$$\eta^{\mathrm{min}} = \frac{\rho_{\mathrm{ER}}^{\mathrm{ideal}}(L)}{\rho_{\mathrm{EW}}^{\infty}} = \frac{\rho_{\mathrm{ER}}^{\infty}}{\rho_{\mathrm{EW}}^{\infty}} \times \frac{L/\lambda_{\mathrm{ER}}}{\sinh(L/\lambda_{\mathrm{ER}})}$$

$$= \eta_{\mathrm{eq}} \times \frac{L/\lambda_{\mathrm{ER}}}{\sinh(L/\lambda_{\mathrm{ER}})} \Rightarrow \tag{S83}$$

$$\frac{\eta^{\mathrm{min}}}{\eta_{\mathrm{eq}}} = \frac{L/\lambda_{\mathrm{ER}}}{\sinh(L/\lambda_{\mathrm{ER}})} = \frac{\sqrt{\tau_D k_{\mathrm{off}}^{\mathrm{R}}}}{\sinh\left(\sqrt{\tau_D k_{\mathrm{off}}^{\mathrm{R}}}\right)}. \tag{S84}$$

Since $L/\lambda_{\mathrm{ER}} < \sinh(L/\lambda_{\mathrm{ER}})$, the lowest fidelity is less than the equilibrium fidelity itself ($\eta^{\mathrm{min}} < \eta_{\mathrm{eq}}$), suggesting that the enzyme may in fact do *anti-proofreading* (*Murugan et al., 2014*) if the wrong substrates are generally closer to the catalytic site.

## 2. Intermediate levels of substrate localization

In *Figure 3* inset as well as in Appendix 1.4, we explored the dependence of fidelity on the substrate localization length scale $\lambda_{\mathrm{S}}$ when it was the same for the two substrate kinds. Here, we expand this study to the case where this constraint is relaxed.

In particular, using *Equation S24*, we calculate complex densities and corresponding fidelity values as a function of $\lambda_{\mathrm{R}}$ for different fixed choices of the length scale ratio $\lambda_{\mathrm{R}}/\lambda_{\mathrm{W}}$. The results of the study are captured in *Appendix 4—figure 1*. In the special case where the two length scales are equal ($\lambda_{\mathrm{R}} = \lambda_{\mathrm{W}}$, solid black line), fidelity exhibits a monotonic depends on $L/\lambda_{\mathrm{R}}$, and in the limit of ideal localization (very large $L/\lambda_{\mathrm{R}}$) the result in *Equation 5* of the main text is recovered.

When $\lambda_{\mathrm{R}} \neq \lambda_{\mathrm{W}}$, the dependence of fidelity on $L/\lambda_{\mathrm{R}}$ is no longer monotonic. If right substrates are more localized than the wrong ones (red curves), then the fidelity curves have a minimum where the enzyme does anti-proofreading (i.e. $\eta < \eta_{\mathrm{eq}}$). The proofreading portion of the curves (when $\eta > \eta_{\mathrm{eq}}$) is shifted to the right, suggesting that much higher substrate localization is needed for the enzyme to proofread.

The opposite case is when the right substrates have a shallower gradient than the wrong ones (blue curves). The fidelity curves are now shifted to the left and have a peak that is greater than the large $L/\lambda_{\mathrm{R}}$ limit of fidelity. This means that there is an optimal degree of substrate localization, going beyond which makes the model performance worse in terms of both error correction and energy consumption.

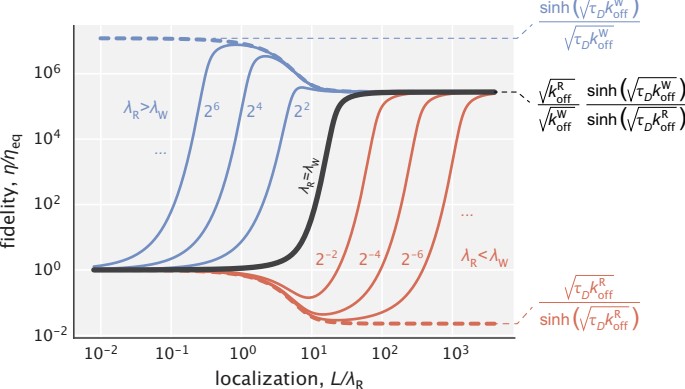

**Appendix 4—figure 1.** Fidelity as a function of $L/\lambda_{\mathrm{R}}$ for different choices of the ratio $\lambda_{\mathrm{R}}/\lambda_{\mathrm{W}}$. The solid black line corresponds to the earlier studied regime where substrates had identical localization length scales. The blue curves represent the cases where $\lambda_{\mathrm{R}} > \lambda_{\mathrm{W}}$, while the red curves represent the cases where $\lambda_{\mathrm{R}} < \lambda_{\mathrm{W}}$. Numbers next to the curves correspond to the $\lambda_{\mathrm{R}}/\lambda_{\mathrm{W}}$ ratios used for generating them. Expressions for the highest and lowest fidelity values, as well as the fidelity expression in the limit where both substrates are highly localized are shown on the right side of the figure. $\tau_D = 40\,\tau_{\mathrm{off}}^{\mathrm{R}}$ and $\eta_{\mathrm{eq}} = 10$ were used for demonstration.

Over the course of its diffusive transport, a bound enzyme is more likely to deposit a right substrate in a substrate-depleted region than a wrong one, because right substrates stay attached to the enzyme for a longer time. Therefore, if the gradient-maintaining mechanism does not discriminate between substrates (which we assume is the case for the kinase/phosphatase-based one), then it will be easier for it to maintain the wrong ones localized since they tend to get deposited closer to the localization site (see *Appendix 6—figure 1c* as an example). This means that in a realistic setting the spatial organization of substrates is more likely to be in the advantageous blue region of *Appendix 4—figure 1* where $\lambda_{\rm R} > \lambda_{\rm W}$, facilitating the realization of spatial proofreading.

## Appendix 5

### Studies on the validity of the uniform free enzyme profile assumption

In our treatment of the model so far, we have assumed for mathematical convenience that free enzymes are in excess, which suggested the approximation $\rho_E(x) \approx \text{constant}$. Example enzyme density profiles shown in *Appendix 5—figure 1*, however, demonstrate that this assumption does not hold in general. Specifically, there is a depletion of free enzymes near the substrate localization site and abundance near the catalysis site. Because of this depletion at the leftmost edge, we expect a reduction in speed in comparison with our earlier treatment where a flat profile was assumed. In addition, if substrates have a weak gradient, we expect the fidelity to also be reduced, since more enzymes will bind substrates at intermediate positions, reducing the average travel distance to the catalytic site. In what follows, we discuss in greater detail the consequences of having a nonuniform free enzyme distribution on the model performance.

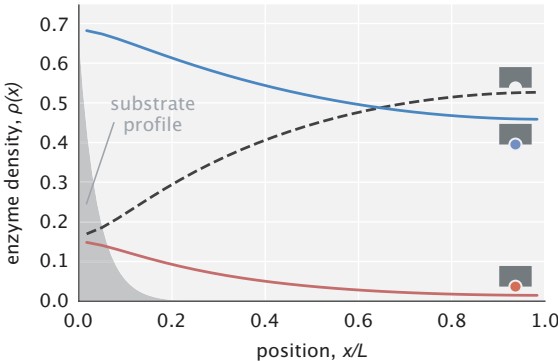

**Appendix 5—figure 1.** Example profiles of free and substrate-bound enzymes. Enzyme profiles are normalized so that the sum of areas under the curves is unity. The substrate profile (rescaled on the *y*-axis) is shown in transparent gray.

### 1. Effects that relaxing the $\rho_E(x) \approx \text{constant}$ assumption has on the Pareto front

We begin by studying the effects of relaxing the uniform free enzyme profile assumption on the Pareto front of the speed–fidelity trade-off (*Figure 3a* of the main text). This front is reached in the ideal substrate localization limit ($\lambda_S \to 0$). Though in general enzyme profiles need to be obtained using numerical methods due to the nonlinearity of reaction–diffusion equations, in this particular limit ($\lambda_S \to 0$) an analytical solution is available. To obtain it, we write the reaction–diffusion equations in the bulk region of space as

$$\frac{\partial \rho_{ER}}{\partial t} = D\frac{\partial^2 \rho_{ER}}{\partial x^2} - k_{off}^{R}\rho_{ER} \tag{S85}$$

$$\frac{\partial \rho_{EW}}{\partial t} = D\frac{\partial^2 \rho_{EW}}{\partial x^2} - k_{off}^{W}\rho_{EW} \tag{S86}$$

$$\frac{\partial \rho_E}{\partial t} = D\frac{\partial^2 \rho_E}{\partial x^2} + \sum_{S=R,W} k_{off}^{S}\rho_{ES}. \tag{S87}$$

Substrate binding reactions did not enter the above equations, as they occur at the leftmost boundary only. They are instead accounted for via boundary conditions, which read

$$-D\frac{\partial \rho_{\mathrm{ER}}}{\partial x}\bigg|_{x=0} = k_{\mathrm{on}}S_{\mathrm{toal}}\rho_{\mathrm{E}}(0), \tag{S88}$$

$$-D\frac{\partial \rho_{\mathrm{EW}}}{\partial x}\bigg|_{x=0} = k_{\mathrm{on}}S_{\mathrm{total}}\rho_{\mathrm{E}}(0), \tag{S89}$$

$$-D\frac{\partial \rho_{\mathrm{E}}}{\partial x}\bigg|_{x=0} = -2k_{\mathrm{on}}S_{\mathrm{total}}\rho_{\mathrm{E}}(0), \tag{S90}$$

where $S_{\mathrm{total}}$ is the total amount of free substrate of each kind concentrated at $x=0$.

## Relating local enzyme concentrations

Considering the system at steady state, we add *Equations S85-S87* and obtain

$$0 = D\frac{\mathrm{d}^2\rho_{\mathrm{ER}}}{\mathrm{d}x^2} + D\frac{\mathrm{d}^2\rho_{\mathrm{EW}}}{\mathrm{d}x^2} + D\frac{\mathrm{d}^2\rho_{\mathrm{E}}}{\mathrm{d}x^2}, \tag{S91}$$

where we replaced the partial derivatives with total derivative since the profiles are time-independent. Dividing *Equation S91* by $D$ and integrating once, we find

$$\frac{\mathrm{d}\rho_{\mathrm{ER}}}{\mathrm{d}x} + \frac{\mathrm{d}\rho_{\mathrm{EW}}}{\mathrm{d}x} + \frac{\mathrm{d}\rho_{\mathrm{E}}}{\mathrm{d}x} = C_1. \tag{S92}$$

The above relation must hold for arbitrary position $x$. Choosing $x=0$ and noting that from *Equations S88-S90* the sum of fluxes should be zero, we can claim that $C_1 = 0$. Integrating for the second time, we obtain

$$\rho_{\mathrm{ER}}(x) + \rho_{\mathrm{EW}}(x) + \rho_{\mathrm{E}}(x) = C_2, \tag{S93}$$

where $C_2$ is now a different constant. To find it, we perform an integral for the last time across the entire compartment, namely,

$$\int_0^L \left( \rho_{\mathrm{ER}}(x) + \rho_{\mathrm{EW}}(x) + \rho_{\mathrm{E}}(x) \right) \mathrm{d}x = E_{\mathrm{total}} = C_2 L. \tag{S94}$$

Here, we introduced the parameter $E_{\mathrm{total}}$ as the total number of enzymes in the system (in free or bound forms). The constant $C_2$, which we will rename into $\rho_0$, is then the average enzyme density, that is,

$$\rho_0 = E_{\mathrm{total}}/L. \tag{S95}$$

Substituting this result into *Equation S93*, we find an insightful relation between free and bound enzyme densities at an arbitrary position, namely,

$$\rho_E(x) = \rho_0 - \rho_{\mathrm{ER}}(x) - \rho_{\mathrm{EW}}(x). \tag{S96}$$

This relation suggests that whenever the local concentration of bound enzymes is high, the local concentration of free enzymes should be correspondingly low, as we see reflected in the profiles of *Appendix 5—figure 1*.

## Deriving the fidelity expression

Next, we consider *Equations S85 and S86* separately at steady state, written in the form

$$D\frac{\mathrm{d}^2\rho_{\mathrm{ES}}}{\mathrm{d}x^2} - k_{\mathrm{off}}^{\mathrm{S}}\rho_{\mathrm{ES}} = 0. \tag{S97}$$

The general solution to this ODE reads

$$\rho_{\mathrm{ES}}(x) = C_1^{\mathrm{S}}e^{-x/\lambda_{\mathrm{ES}}} + C_2^{\mathrm{S}}e^{x/\lambda_{\mathrm{ES}}}, \tag{S98}$$

where $\lambda_{\mathrm{ES}} = \sqrt{D/k_{\mathrm{off}}^{\mathrm{S}}}$, and $C_1^{\mathrm{S}}$ and $C_2^{\mathrm{S}}$ (S = R,W) are constants which are different for right and wrong

complexes. The no-flux boundary condition at $x=L$ can be used to relate these constants and simplify the complex profile expression, namely,

$$-D\frac{d\rho_{ES}(x)}{dx}\Big|_{x=L}=-\frac{D}{\lambda_{ES}}\left(-C_1^S e^{-L/\lambda_{ES}}+C_2^S e^{L/\lambda_{ES}}\right)=0\Rightarrow \tag{S99}$$

$$C_2^S=e^{-2L/\lambda_{ES}}C_1^S\Rightarrow \tag{S100}$$

$$\rho_{ES}(x)=C_1^S e^{-x/\lambda_{ES}}+C_1^S e^{-2L/\lambda_{ES}}e^{x/\lambda_{ES}}$$

$$=2C_1^S e^{-L/\lambda_{ES}}\cosh\left(\frac{L-x}{\lambda_{ES}}\right)$$

$$=\tilde{C}_1^S\cosh\left(\frac{L-x}{\lambda_{ES}}\right), \tag{S101}$$

where $\tilde{C}_1^S=2C_1^S e^{-L/\lambda_{ES}}$ is a new constant coefficient introduced for convenience.

Now, the fidelity of the scheme is the ratio of right and wrong complex densities at $x=L$. Using the result above, the fidelity can be written as

$$\eta=\frac{\rho_{ER}(L)}{\rho_{EW}(L)}=\frac{\tilde{C}_1^R}{\tilde{C}_1^W}. \tag{S102}$$

The ratio of these constant coefficients can be obtained by noting that the diffusive fluxes of right and wrong complexes at $x=0$ are identical (from *Equations S38 and S38*), that is,

$$-D\frac{\partial\rho_{ER}}{\partial x}\Big|_{x=0}=-D\frac{\partial\rho_{EW}}{\partial x}\Big|_{x=0}\Rightarrow \tag{S103}$$

$$\tilde{C}_1^R\times\frac{\sinh(L/\lambda_{ER})}{\lambda_{ER}}=\tilde{C}_1^W\times\frac{\sinh(L/\lambda_{EW})}{\lambda_{EW}}\Rightarrow \tag{S104}$$

$$\frac{\tilde{C}_1^R}{\tilde{C}_1^W}=\frac{\lambda_{ER}}{\lambda_{EW}}\frac{\sinh(L/\lambda_{EW})}{\sinh(L/\lambda_{ER})}. \tag{S105}$$

Substituting this result into *Equation S102*, and recalling the equality $L/\lambda_{ES}=\sqrt{\tau_D k_{off}^S}$, we obtain

$$\eta=\frac{\sqrt{\tau_D k_{off}^W}}{\sqrt{\tau_D k_{off}^R}}\frac{\sinh\left(\sqrt{\tau_D k_{off}^W}\right)}{\sinh\left(\sqrt{\tau_D k_{off}^R}\right)}=\sqrt{\eta_{eq}}\frac{\sinh\left(\sqrt{\tau_D k_{off}^W}\right)}{\sinh\left(\sqrt{\tau_D k_{off}^R}\right)}. \tag{S106}$$

This expression is identical to that in *Equation S20* which was derived under the $\rho_E(x)\approx\text{constant}$ assumption, suggesting that when substrates are highly localized, the shape of the free enzyme profile does not dictate the fidelity.

## Deriving the speed expression

To keep the expression of speed compact while still illustrating the key consequences of relaxing the $\rho(x)\approx\text{constant}$ assumption, we will assume moving forward that the density of wrong complexes is much lower than that of the right complexes, that is, $\rho_{EW}(x)\ll\rho_{ER}(x)$. This assumption holds as long as the right and wrong complexes have sufficiently different off-rates. To see why it is the case, note that the ratio $\rho_{EW}(x)/\rho_{ER}(x)$ is the highest at $x=0$. We therefore calculate an upper bound for the ratio using *Equation S101* and *Equation S105* as

$$\frac{\rho_{EW}(x)}{\rho_{ER}(x)}<\frac{\rho_{EW}(0)}{\rho_{ER}(0)}=\frac{\lambda_{EW}}{\lambda_{ER}}\frac{\tanh(L/\lambda_{ER})}{\tanh(L/\lambda_{EW})}<\frac{\lambda_{EW}}{\lambda_{ER}}=\sqrt{\frac{k_{off}^R}{k_{off}^W}}=\frac{1}{\sqrt{\eta_{eq}}}. \tag{S107}$$

As long as $\eta_{eq}\gtrsim 10$, it is fair to assume that the right complexes greatly outnumber the wrong ones, which allows us to approximate the free enzyme density from *Equation S96* as $\rho_E(x)\approx\rho_0-\rho_{ER}(x)$.

The specification of the right complex density profile requires the knowledge of the unknown coefficient $\tilde{C}_1^R$. To find this coefficient, we use the boundary condition in *Equation S88* and the approximation $\rho_E(x)\approx\rho_0-\rho_{ER}(x)$ to write

$$D\frac{\tilde{C}_1^R}{\lambda_{ER}}\sinh(L/\lambda_{ER}) = k_{on}S_{total}\left(\rho_0 - \tilde{C}_1^R\cosh(L/\lambda_{ER})\right) \Rightarrow \tag{S108}$$

$$
\begin{aligned}
\tilde{C}_1^R &= \frac{k_{on}S_{total}\rho_0}{\frac{D}{\lambda_{ER}}\sinh(L/\lambda_{ER}) + k_{on}S_{total}\cosh(L/\lambda_{ER})} \\
&= \frac{k_{on}S_{total}\rho_0}{\lambda_{ER}k_{off}^R\sinh(L/\lambda_{ER}) + k_{on}S_{total}\cosh(L/\lambda_{ER})} \\
&= \rho_0 \times \frac{\frac{k_{on}S_{total}}{k_{off}^R L}}{1 + \frac{L}{\lambda_{ER}}\frac{\cosh(L/\lambda_{ER})}{\sinh(L/\lambda_{ER})}\frac{k_{on}S_{total}}{k_{off}^R L}} \times \frac{L/\lambda_{ER}}{\sinh(L/\lambda_{ER})}.
\end{aligned}
\tag{S109}
$$

With the constant coefficient known, the right complex density then becomes

$$\rho_{ER}(x) = \rho_0 \times \frac{\frac{\bar{\rho}_S}{K_d^R}}{1 + \frac{L}{\lambda_{ER}}\frac{\cosh(L/\lambda_{ER})}{\sinh(L/\lambda_{ER})}\frac{\bar{\rho}_S}{K_d^R}} \times \frac{L/\lambda_{ER}}{\sinh(L/\lambda_{ER})}\cosh\left(\frac{L-x}{\lambda_{ER}}\right), \tag{S110}$$

where we used the definitions of the mean substrate density $\bar{\rho}_S = S_{total}/L$ and the dissociation constant $K_d^R = k_{off}^R/k_{on}$.

To enable a direct parallel between this general treatment and the earlier one with the $\rho_E(x) \approx$ constant approximation, let us introduce $\rho_{ER}^\infty$ as the uniform right complex density when diffusion is very fast ($\lambda_{ER} \gg L$) and calculate it from *Equation S110* as

$$\rho_{ER}^\infty = \rho_0 \times \frac{\frac{\bar{\rho}_S}{K_d^R}}{1 + \frac{\bar{\rho}_S}{K_d^R}}. \tag{S111}$$

Now, using the $\rho_{ER}^\infty$ expression, we rewrite *Equation S110* as

$$
\begin{aligned}
\rho_{ER}(x) &= \frac{1 + \frac{\bar{\rho}_S}{K_d^R}}{1 + \frac{L}{\lambda_{ER}}\frac{\cosh(L/\lambda_{ER})}{\sinh(L/\lambda_{ER})}\frac{\bar{\rho}_S}{K_d^R}} \times \rho_{ER}^\infty \times \frac{L/\lambda_{ER}}{\sinh(L/\lambda_{ER})}\cosh\left(\frac{L-x}{\lambda_{ER}}\right) \\
&= \frac{1 + \frac{\bar{\rho}_S}{K_d^R}}{\underbrace{1 + \frac{L}{\lambda_{ER}}\frac{\cosh(L/\lambda_{ER})}{\sinh(L/\lambda_{ER})}\frac{\bar{\rho}_S}{K_d^R}}_{\gamma}} \times \rho_{ER}^{const}(x),
\end{aligned}
\tag{S112}
$$

where $\rho_{ER}^{const}(x)$ is the complex density obtained under the $\rho_E(x) \approx$ constant assumption (*Equation S15*). The extra factor that appears on front does not exceed 1 since $\gamma \geq 1$, indicating a reduction in speed, as we anticipated in our more qualitative discussion at the beginning of the section. The presence of the extra factor suggests two possibilities for the approximation to hold true; first, $\gamma \approx 1$ which happens when $\lambda_{ER} \gtrsim L$ or when the right complex does not decay noticeably across the compartment, and second, when $\gamma > 1$ and $\bar{\rho}_S \ll \gamma^{-1}K_d^R$, which is when right complexes do decay but their fraction is low compared with free enzymes because of low substrate concentration.

Let us demonstrate the last statement more explicitly. Specifically, let us show that the validity of the approximation $\rho_E(x) \approx$ constant is indeed linked directly to the fraction of bound enzymes. To that end, we evaluate $\rho_E(0)/\rho_E(L)$ as a metric that quantifies the degree to which $\rho_E(x) \approx$ constant holds. If there is a large depletion of free enzymes near the substrate-binding site, then the metric will be significantly less than 1; conversely, if the free enzyme profile is practically flat, then the metric will be close to 1. Invoking the relation $\rho_E(x) \approx \rho_0 - \rho_{ER}(x)$ and using our result for the complex density (*Equation S110*) as well as the definition of $\gamma$ in *Equation S112*, we evaluate this metric as

$$
\begin{aligned}
\frac{\rho_{\mathrm{E}}(0)}{\rho_{\mathrm{E}}(L)} &\approx \frac{\rho_0 - \rho_{\mathrm{ER}}(0)}{\rho_0 - \rho_{\mathrm{ER}}(L)} \\
&= \frac{1 - \frac{\gamma\bar{\rho}_{\mathrm{S}}/K_{\mathrm{d}}^{\mathrm{R}}}{1+\gamma\bar{\rho}_{\mathrm{S}}/K_{\mathrm{d}}^{\mathrm{R}}}}{1 - \frac{\gamma\bar{\rho}_{\mathrm{S}}/K_{\mathrm{d}}^{\mathrm{R}}}{\cosh(L/\lambda_{\mathrm{ER}})(1+\gamma\bar{\rho}_{\mathrm{S}}/K_{\mathrm{d}}^{\mathrm{R}})}} \\
&= \frac{1}{1 + \left(1 - \frac{1}{\cosh(L/\lambda_{\mathrm{ER}})}\right)\gamma\bar{\rho}_{\mathrm{S}}/K_{\mathrm{d}}^{\mathrm{R}}} .
\end{aligned}
\tag{S113}
$$

Next, we calculate the fraction of bound enzymes $p_{\mathrm{bound}}$ from *Equation S110* as

$$
\begin{aligned}
p_{\mathrm{bound}} &\approx E_{\mathrm{total}}^{-1} \int_0^L \rho_{\mathrm{ER}}(x)\,\mathrm{d}x \\
&= \frac{\rho_0 L}{E_{\mathrm{total}}} \frac{\bar{\rho}_{\mathrm{S}}/K_{\mathrm{d}}^{\mathrm{R}}}{1 + \gamma\bar{\rho}_{\mathrm{S}}/K_{\mathrm{d}}^{\mathrm{R}}} \\
&= \frac{\bar{\rho}_{\mathrm{S}}/K_{\mathrm{d}}^{\mathrm{R}}}{1 + \gamma\bar{\rho}_{\mathrm{S}}/K_{\mathrm{d}}^{\mathrm{R}}} .
\end{aligned}
\tag{S114}
$$

Note that $\gamma^{-1}$ emerges as the highest fraction of bound enzymes ($p_{\mathrm{bound}}^{\mathrm{max}}$) reached in the large substrate concentration limit.

To link the metric $\rho_{\mathrm{E}}(0)/\rho_{\mathrm{E}}(L)$ to the fraction of bound enzymes, we express $\bar{\rho}_{\mathrm{S}}/K_{\mathrm{d}}^{\mathrm{R}}$ in terms of $p_{\mathrm{bound}}$ and substitute it into *Equation S113*, namely,

$$
\bar{\rho}_{\mathrm{S}}/K_{\mathrm{d}}^{\mathrm{R}} = \frac{p_{\mathrm{bound}}}{1 - \gamma p_{\mathrm{bound}}} \Rightarrow
\tag{S115}
$$

$$
\begin{aligned}
\frac{\rho_{\mathrm{E}}(0)}{\rho_{\mathrm{E}}(L)} &= \frac{1}{1 + \left(1 - \frac{1}{\cosh(L/\lambda_{\mathrm{ER}})}\right)\frac{\gamma p_{\mathrm{bound}}}{1 - \gamma p_{\mathrm{bound}}}} \\
&= \frac{1 - \gamma p_{\mathrm{bound}}}{(1 - \gamma p_{\mathrm{bound}}) + \left(1 - \frac{1}{\cosh(L/\lambda_{\mathrm{ER}})}\right)\gamma p_{\mathrm{bound}}} \\
&= \frac{1 - \gamma p_{\mathrm{bound}}}{1 - \gamma p_{\mathrm{bound}}/\cosh(L/\lambda_{\mathrm{ER}})} \\
&= \frac{p_{\mathrm{bound}}^{\mathrm{max}} - p_{\mathrm{bound}}}{p_{\mathrm{bound}}^{\mathrm{max}} - p_{\mathrm{bound}}/\cosh(L/\lambda_{\mathrm{ER}})} .
\end{aligned}
\tag{S116}
$$

Now, when the complexes do not decay appreciably across the compartment ($\lambda_{\mathrm{ER}} \gtrsim L$ and thus, $\cosh(L/\lambda_{\mathrm{ER}}) \approx 1$), the metric becomes roughly equal to 1, suggesting that the free enzyme profile is practically flat. A more interesting case is when the complexes do decay ($\lambda_{\mathrm{ER}} < L$), as in *Appendix 5—figure 1*. In this case, applying the condition $\cosh(L/\lambda_{\mathrm{ER}}) \gg 1$, we find

$$
\frac{\rho_{\mathrm{E}}(0)}{\rho_{\mathrm{E}}(L)} \approx 1 - \frac{p_{\mathrm{bound}}}{p_{\mathrm{bound}}^{\mathrm{max}}} .
\tag{S117}
$$

The anti-correlation between the $\rho_{\mathrm{E}}(0)/\rho_{\mathrm{E}}(L)$ and $p_{\mathrm{bound}}$ in the above result demonstrates that the degree to which the approximation $\rho_{\mathrm{E}}(x) \approx \mathrm{constant}$ is violated is indeed dictated by the fraction of bound enzymes.

### Pareto front shift

The previous calculations showed that in the ideal substrate localization limit relaxing the $\rho(x) \approx \mathrm{constant}$ assumption keeps the fidelity the same while the speed gets reduced. And this reduction is greater for higher substrate concentrations. We therefore expect a shift in the Pareto front when going to the high substrate concentration limit, as is illustrated in *Appendix 5—figure 2a*. To get more intuition about the effect of this shift caused by tuning the amount of substrates, we consider the effective number of proofreading realizations at half-maximum speed ($n_{50}$) and study how this number changes as a function of the fraction of enzymes bound ($p_{\mathrm{bound}}$), which increases monotonically with $S_{\mathrm{total}}$ as suggested by *Equation S114*. *Appendix 5—figure 2b* shows this dependence. As can be seen, $n_{50}$ reduces roughly linearly with $p_{\mathrm{bound}}$; for example, if 10% of the enzymes are

bound, then a 10% reduction in $n_{50}$ is expected. This suggests that as long as the fraction of bound enzymes is low, our findings related to the Pareto front made under the $\rho_E \approx \text{constant}$ assumption will generally hold true.

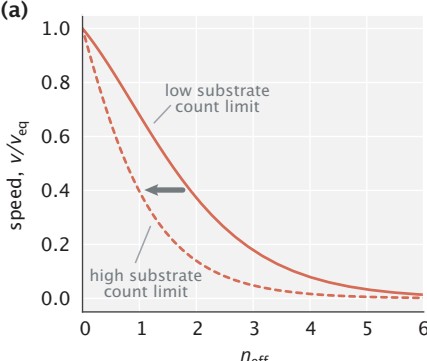 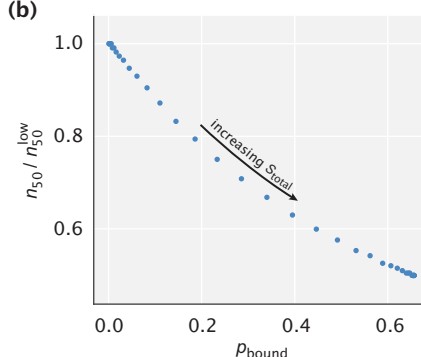

**Appendix 5—figure 2.** Consequences of relaxing the $\rho_E(x) \approx \text{constant}$ assumption on the Pareto front. (a) Pareto fronts in the low and high substrate concentration limits. (b) Reduction in the effective number of proofreading realizations at half-maximum speed as a function of the fraction of enzymes bound. $\eta_{eq} = 10$ was used in making the plots.

## 2. Effects that relaxing the $\rho_E(x) \approx \text{constant}$ assumption has on fidelity in a weak substrate gradient setting

In this section, we study how accounting for the spatial distribution of free enzymes affects our results on the model's fidelity in the setting where substrates have a finite localization length scale $\lambda_s$. In this setting, *Equations (1–3)* (in the main text) describing the system's dynamics become a system of nonlinear equations, which we solve at steady state using numerical methods.

An example curve of how fidelity changes with tuning diffusion time scale in a finite $\lambda_s$ setting is shown in *Appendix 5—figure 3*. As expected, the nonuniform free enzyme profile leads to a reduction in fidelity. This reduction is not significant when diffusion is relatively fast as in that case the free enzyme profile manages to flatten out rapidly. The reduction is not significant also in the very slow diffusion limit where binding events that lead to production primarily take place in the proximity of the activation region and hence, the nonuniform profile of free enzymes across the compartment has little impact on fidelity. The greatest reduction happens at intermediate diffusion time scales; in particular, when the system achieves its peak fidelity.

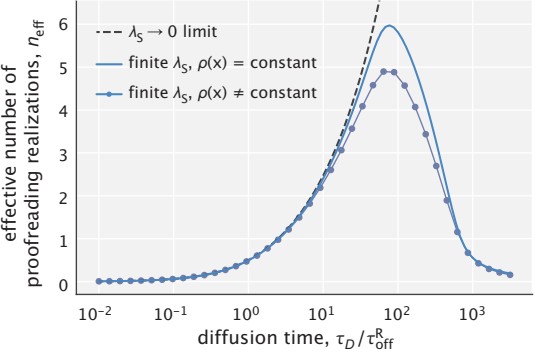

**Appendix 5—figure 3.** Fidelity as a function of diffusion time scale calculated with and without making the $\rho_E(x) \approx \text{constant}$ approximation. The total number of free substrates is chosen so that $\bar{\rho}_s/K_d^R = 3$. The substrate localization length scale used for generating the solid curves is $\lambda_s/L = 0.04$.

To quantify the extent of this highest reduction, we calculated the peak value of the effective number of proofreading realizations ($n_{\max}$) for different free substrate amounts which regulate the fraction of bound enzymes ($p_{\text{bound}}$). The results obtained for different choices of $\lambda_{\text{s}}$ are summarized in *Appendix 5—figure 4*. As can be seen, for the high substrate localization case ($\lambda_{\text{s}}/L = 0.04$), there is a roughly linear dependence between $n_{\max}$ and $p_{\text{bound}}$. The initial decrease in $n_{\max}$ with growing $p_{\text{bound}}$ is even slower when substrates are less tightly localized ($\lambda_{\text{s}}/L = 0.10, 0.30$).

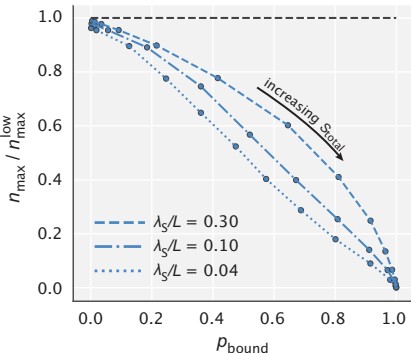

**Appendix 5—figure 4.** Reduction in the peak effective number of proofreading realizations as a function of $p_{\text{bound}}$. $n_{\max}^{\text{low}}$ represents the peak value of $n_{\text{eff}}$ in the limit of low substrate concentration (the maximum of the solid blue curve in *Appendix 5—figure 3*).

Taken together, these results suggest that if the substrate concentration is low enough to leave most of the enzymes unbound, then our proposed scheme will proofread efficiently. And this requirement on substrate amount will be further relaxed if diffusion is fast, or if substrates are not very tightly localized.

## Appendix 6

## Proofreading on a kinase/phosphatase-induced gradient

In this section, we introduce the mathematical modeling setup for the kinase/phosphatase-based gradient formation scheme and describe how its fidelity is calculated numerically. In the end, we discuss the energetics of setting up the substrate concentration gradient and link our calculations to the lower bounds on energy cost obtained earlier in Appendix 2.

### 1. Setup and estimation of fidelity

In the analysis thus far, we have imposed a gradient of free substrates and analyzed the proofreading capability of an enzyme acting on this gradient. In a living cell, gradients themselves are maintained by active cellular processes. However, the action of the enzyme – that is, binding a substrate in one spatial location, diffusing away, and releasing the substrate elsewhere – can destroy the gradient, and thereby lead to a loss of proofreading. Here, we analyze the consequences of free substrate depletion and gradient flattening caused by the enzyme.

We model the formation of a substrate gradient by a combination of localized activation and delocalized deactivation. We suppose that substrates can exist in phosphorylated or dephosphorylated forms, and that only the phosphorylated form is capable of binding to the enzyme. The substrates are phosphorylated by a kinase with rate $k_{\mathrm{kin}} = 0.2$ s$^{-1}$, and dephosphorylated by a phosphatase with rate $k_{\mathrm{p}} = 5$ s$^{-1}$. Crucially, we assume that phosphatases are found everywhere in the domain of size $L \sim 10$ $\mu$m (a typical length scale in a eukaryotic cell), while kinases are localized to one end of the domain (at $x = 0$), as may occur naturally if kinases are bound to one of the membranes enclosing the domain.

The minimal dynamics of phosphorylated substrates and enzyme–substrate complexes is then given by

$$\begin{aligned}
\frac{\partial \rho_{\mathrm{S}}}{\partial t} &= D\nabla^2 \rho_{\mathrm{S}} - k_{\mathrm{b}}\rho_{\mathrm{S}} + k_{\mathrm{off}}^{\mathrm{S}}\rho_{\mathrm{ES}} - k_{\mathrm{p}}\rho_{\mathrm{S}}, \\
\frac{\partial \rho_{\mathrm{ES}}}{\partial t} &= D\nabla^2 \rho_{\mathrm{ES}} + k_{\mathrm{b}}\rho_{\mathrm{S}} - k_{\mathrm{off}}^{\mathrm{S}}\rho_{\mathrm{ES}},
\end{aligned}$$

(S118)

augmented by the boundary conditions

$$\begin{aligned}
&\mathrm{Substrate\,phosphorylation:} -D\nabla\rho_{\mathrm{S}}|_{x=0} = k_{\mathrm{kin}}, \\
&\mathrm{No-flux:} -D\nabla\rho_{\mathrm{S}}|_{x=L} = -D\nabla\rho_{\mathrm{ES}}|_{x=L} = -D\nabla\rho_{\mathrm{ES}}|_{x=0} = 0.
\end{aligned}$$

(S119)

Here, we have supposed that the densities of free enzymes, dephosphorylated substrates, and phosphatases are fixed and uniform, and have absorbed them into the relevant rate constants ($k_{\mathrm{b}} = k_{\mathrm{on}}\rho_{\mathrm{E}}$, $k_{\mathrm{kin}}$, and $k_{\mathrm{p}}$, respectively). For simplicity, we have also assumed that the free substrates and enzyme–substrate complexes have the same diffusion coefficient $D = 1$ $\mu$m$^2$/s. We note that accounting for distinct diffusivities of phosphorylated and unphosphorylated substrate forms (*Kholodenko, 2009*) would affect the speed, while accounting for the slower diffusion of the enzyme–substrate complex would alter the estimates of both speed and fidelity of the model. One or several of these effects can be considered when studying a specific biological system where these microscopic details are known.

We numerically solve *Equations S118 and S119* at steady state to obtain the concentration profiles. First, the equations of dynamics are made dimensionless by settings units of length and time by $L$ ($\bar{x} = x/L$) and $\tau_D \equiv L^2/D$ ($\bar{t} = t/\tau_D$), respectively. At steady state, the dimensionless equations read

$$\begin{aligned}
\bar{\nabla}^2 \bar{\rho}_{\mathrm{S}} &= (\bar{k}_{\mathrm{b}} + \bar{k}_{\mathrm{p}})\bar{\rho}_{\mathrm{S}} - \bar{k}_{\mathrm{off}}^{\mathrm{S}}\bar{\rho}_{\mathrm{ES}}, \\
\bar{\nabla}^2 \bar{\rho}_{\mathrm{ES}} &= -\bar{k}_{\mathrm{b}}\bar{\rho}_{\mathrm{S}} + k_{\mathrm{off}}^{\mathrm{S}}\bar{\rho}_{\mathrm{ES}},
\end{aligned}$$

(S120)

with boundary conditions

$$\begin{aligned}
\bar{\nabla}\bar{\rho}_{\text{S}}\big|_{\bar{x}=0} &= -\bar{k}_{\text{kin}}, \\
\bar{\nabla}\bar{\rho}_{\text{S}}\big|_{\bar{x}=1} &= \bar{\nabla}\bar{\rho}_{\text{ES}}\big|_{\bar{x}=1} = \bar{\nabla}\bar{\rho}_{\text{ES}}\big|_{\bar{x}=0} = 0,
\end{aligned} \tag{S121}$$

where concentrations have been rescaled as $\bar{\rho} = \rho L$, and kinetic rates as $\bar{k} = k\,\tau_D$.

We discretize the steady state equations on a grid with spacing $\Delta\bar{x} = 0.01$, approximating the second derivative as

$$\bar{\nabla}^2\bar{\rho} \approx \frac{1}{\Delta\bar{x}^2}\left(\bar{\rho}(\bar{x}+\Delta\bar{x}) + \bar{\rho}(\bar{x}-\Delta\bar{x}) - 2\bar{\rho}(\bar{x})\right). \tag{S122}$$

This is ill-defined at the boundaries $\bar{x}=0$ and $\bar{x}=1$, which is addressed by incorporating the boundary conditions. For illustration, consider the left boundary, $\bar{x}=0$, and suppose that our domain included also a point at $\bar{x}=-\Delta\bar{x}$. Then, we could approximate the boundary condition $\bar{\nabla}\bar{\rho}_{\text{S}}\big|_{\bar{x}=0} = -\bar{k}_{\text{kin}}$ by a centred difference scheme, and solve out for the fictional point at $\bar{x}=-\Delta\bar{x}$, namely,

$$\begin{aligned}
&\bar{\nabla}\bar{\rho}_{\text{S}}\big|_{\bar{x}=0} = -\bar{k}_{\text{kin}} \\
\Rightarrow &\frac{1}{2\Delta\bar{x}}\left(\bar{\rho}_{\text{S}}(\Delta\bar{x}) - \bar{\rho}_{\text{S}}(-\Delta\bar{x})\right) = -\bar{k}_{\text{kin}} \\
\Rightarrow &\bar{\rho}_{\text{S}}(-\Delta\bar{x}) = \bar{\rho}_{\text{S}}(\Delta\bar{x}) + 2\Delta\bar{x}\,\bar{k}_{\text{kin}},
\end{aligned}$$

which, when inserted into *Equation S122*, specifies $\bar{\nabla}^2\bar{\rho}_{\text{S}}$ at $\bar{x}=0$, that is,

$$\bar{\nabla}^2\bar{\rho}_{\text{S}}\big|_{\bar{x}=0} = \frac{1}{\Delta\bar{x}^2}\left(2\bar{\rho}_{\text{S}}(\Delta\bar{x}) - 2\bar{\rho}_{\text{S}}(0)\right) + \frac{2}{\Delta\bar{x}}\bar{k}_{\text{kin}}. \tag{S123}$$

For the boundary at the right ($\bar{x}=1$) as well as for the boundary conditions for $\bar{\rho}_{\text{ES}}$, we similarly implement no-flux boundary conditions. After discretizing, *Equation S120* can then be written in a matrix form as

$$\begin{aligned}
\overbrace{\left(\frac{1}{\Delta\bar{x}^2}\begin{pmatrix} -2 & 2 & 0 & \cdots & 0 \\ 1 & -2 & 1 & \cdots & 0 \\ \vdots & \vdots & \vdots & \ddots & \vdots \\ 0 & \cdots & 1 & -2 & 1 \\ 0 & 0 & \cdots & 1 & -1 \end{pmatrix} - (\bar{k}_{\text{b}} + \bar{k}_{\text{p}})\mathbf{I}\right)}^{\mathbf{M}_{\text{S}}} \vec{\rho}_{\text{S}} &= -\bar{k}_{\text{off}}^{\text{S}}\vec{\rho}_{\text{ES}} + \overbrace{\begin{pmatrix} -\frac{2}{\Delta\bar{x}}\bar{k}_{\text{kin}} \\ 0 \\ \vdots \\ 0 \\ 0 \end{pmatrix}}^{\vec{b}}, \\
\underbrace{\left(\frac{1}{\Delta\bar{x}^2}\begin{pmatrix} -1 & 1 & 0 & \cdots & 0 \\ 1 & -2 & 1 & \cdots & 0 \\ \vdots & \vdots & \vdots & \ddots & \vdots \\ 0 & \cdots & 1 & -2 & 1 \\ 0 & 0 & \cdots & 1 & -1 \end{pmatrix} - \bar{k}_{\text{off}}^{\text{S}}\mathbf{I}\right)}_{\mathbf{M}_{\text{ES}}} \vec{\rho}_{\text{ES}} &= -\bar{k}_{\text{b}}\vec{\rho}_{\text{S}},
\end{aligned} \tag{S124}$$

where $\vec{\rho}_{\text{S}}$, $\vec{\rho}_{\text{ES}}$ are column vectors of the nondimensionalized concentration profiles evaluated at the spatial grid points, that is, $[\bar{\rho}(0), \bar{\rho}(\Delta\bar{x}), \cdots]^T$. Solving these matrix equations yields

$$\begin{aligned}
\vec{\rho}_{\text{S}} &= \left(\mathbf{M}_{\text{S}} - \bar{k}_{\text{off}}^{\text{S}}\bar{k}_{\text{b}}\mathbf{M}_{\text{ES}}^{-1}\right)^{-1}\vec{b}, \\
\vec{\rho}_{\text{ES}} &= -\bar{k}_{\text{b}}\left(\mathbf{M}_{\text{S}}\mathbf{M}_{\text{ES}} - \bar{k}_{\text{off}}^{\text{S}}\bar{k}_{\text{b}}\mathbf{I}\right)^{-1}\vec{b}.
\end{aligned} \tag{S125}$$

We compute *Equation S125* numerically for two substrates: a cognate ('R') and a non-cognate ('W'), which differ in their off-rates ($k_{\text{off}}^{\text{R}} = 0.1\,\text{s}^{-1}$ and $k_{\text{off}}^{\text{W}} = 1\,\text{s}^{-1}$, respectively). Having the density profiles, the fidelity of the model becomes $\eta \approx \bar{\rho}_{\text{ER}}(\bar{x}=1)/\bar{\rho}_{\text{EW}}(\bar{x}=1)$. We calculate the fidelity for different choices of the first–order rate of enzyme–substrate binding ($k_{\text{b}} = k_{\text{on}}\rho_{\text{E}}$); this may be thought of as varying the concentration of free enzyme in the cell. The results are shown in *Figure 5* of the main text.

## 2. Energy dissipation

In Appendices 2.1 and 2.3, we estimated lower bounds on the minimum power that needs to be dissipated in order to counter the homogenizing effect that enzyme activity and substrate diffusion respectively have on localized substrate profiles. Here, we calculate the energy dissipation required to run the kinase/phosphatase-based mechanism and compare it with these lower bounds estimated earlier.

Let us assume that phosphorylation and dephosphorylation reactions by kinases and phosphatases are nearly irreversible with associated free energy costs of $\Delta\varepsilon_{\text{kin}}$ and $\Delta\varepsilon_{\text{phosph}}$ per reaction, respectively. The net rate at which active substrates get dephosphorylated is $k_{\text{p}}S_{\text{phosphorylated}}$ and it needs to be identical to the net phosphorylation rate of inactive substrates in order for $S_{\text{phosphorylated}}$ to remain constant. With the costs of each reaction known, we can write the rate of energy dissipation $P_{\text{k/p}}$ as

$$P_{\text{k/p}} = k_{\text{p}}S_{\text{phosphorylated}}(\Delta\varepsilon_{\text{kin}} + \Delta\varepsilon_{\text{phosph}}). \tag{S126}$$

To gain analytical intuition, we first consider the case where the enzyme activity is very low, so that the kinase/phosphatase–based mechanism maintains an exponential profile of active substrates with a decay length scale $\lambda_{\text{s}} = \sqrt{D_{\text{s}}/k_{\text{p}}}$. Expressing the rate of phosphorylation in terms of $\lambda_{\text{s}}$ and $D_{\text{s}}$ (i.e., $k_{\text{p}} = D_{\text{s}}/\lambda_{\text{s}}^2$), and substituting it into *Equation S126*, we obtain

$$P_{\text{k/p}} = \frac{D_{\text{s}}S_{\text{phosphorylated}}}{\lambda_{\text{s}}^2}(\Delta\varepsilon_{\text{kin}} + \Delta\varepsilon_{\text{phosph}}). \tag{S127}$$

Comparing this result with the lower dissipation bound found earlier (*Equation S43*), we can note the presence of an extra factor $\beta(\Delta\varepsilon_{\text{kin}} + \Delta\varepsilon_{\text{phosph}})$. Since the free energy consumption during ATP hydrolysis is $\sim 10\,k_{\text{B}}T$, we can say that the power dissipated by the kinase/phosphatase system for setting up an exponential gradient surpasses the lower limit necessary for countering diffusion roughly by an order of magnitude.

Next, we explore the energetics of the kinase/phosphatase-based mechanism in the context of the power–fidelity trade-off. Our study of the trade-off in *Figure 4* of the main text was performed under the assumption that substrate profiles were exponentially decaying in the entire spatial domain. In *Appendix 6—figure 1a*, we show the trade-off curves obtained under this assumption and compare them with the trade-off curve for the kinase/phosphatase-based mechanism that arises in response to changing the substrate localization by tuning $k_{\text{p}}$. As can be seen, the predicted lower bound (sum of the minimum powers needed to counteract the enzyme action and substrate diffusion) is roughly an order of magnitude lower than the total dissipation of the mechanism, and this difference increases with higher fidelity.

Note, however, that the assumption about an exponential substrate localization is not generally valid for the kinase/phosphatase-based mechanism because substrates can be deposited in low–concentration regions and not get immediately dephosphorylated (*Appendix 6—figure 1c*). We therefore refine our lower bounds on the dissipated power by estimating them numerically using their generic definitions, namely, *Equation S30* for counteracting enzymatic action, and *Equation S42* for counteracting substrate diffusion. These refined estimates suggest a factor of ~10 difference between the total cost and its lower bound consistently across a wide region of the trade-off curve. This means that substrate gradient maintenance through practically irreversible phosphorylation and dephosphorylation reactions has low energetic efficiency for doing spatial proofreading, which, however, may be sustainable depending on the energy budget of the cell.

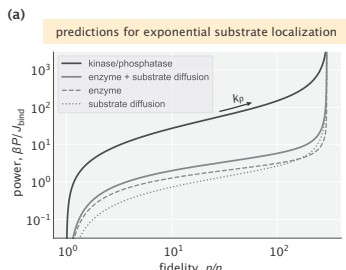

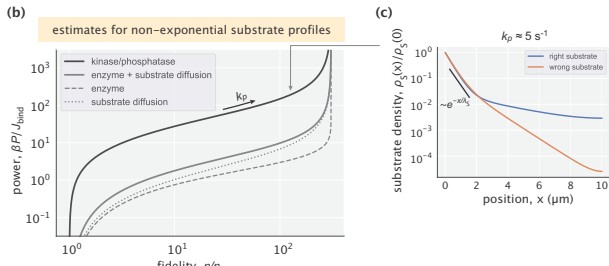

**Appendix 6—figure 1.** Energetic performance of the kinase/phosphatase–based mechanism. (a) Total dissipation and calculated lower bounds under the assumption of exponential substrate localization. (b) Total dissipation and lower bounds estimated without assuming exponential substrate profiles. In both (a) and (b) $k_b = 1\,\mathrm{s}^{-1}$ and $\Delta\varepsilon_{\mathrm{kin}} = \Delta\varepsilon_{\mathrm{phosph}} = 10\,k_{\mathrm{B}}T$ were used. (c) Example profiles of right and wrong substrates for the physiologically relevant dephosphorylation rate $k_p = 5\,\mathrm{s}^{-1}$. Exponential decay of the substrate profile with the predicted length scale $\lambda_{\mathrm{s}} = \sqrt{D_{\mathrm{s}}/k_p}$ holds in the first $\sim 2\,\mu\mathrm{m}$ of the compartment.

