## [Decision Letter]

Thank you for submitting your article "Proofreading through spatial gradients" for consideration by *eLife*. Your article has been reviewed by three peer reviewers, and the evaluation has been overseen by a Reviewing Editor and Aleksandra Walczak as the Senior Editor. The reviewers have opted to remain anonymous.

The reviewers have discussed the reviews with one another and the Reviewing Editor has drafted this decision to help you prepare a revised submission.

As the editors have judged that your manuscript is of interest, but as described below that substantial revisions are required before it is published, we would like to draw your attention to changes in our revision policy that we have made in response to COVID-19 (https://elifesciences.org/articles/57162). First, because many researchers have temporarily lost access to the labs, we will give authors as much time as they need to submit revised manuscripts. We are also offering, if you choose, to post the manuscript to bioRxiv (if it is not already there) along with this decision letter and a formal designation that the manuscript is "in revision at *eLife*". Please let us know if you would like to pursue this option. (If your work is more suitable for medRxiv, you will need to post the preprint yourself, as the mechanisms for us to do so are still in development.)

Summary:

In the manuscript by Galstyn et al. on "Proofreading through spatial gradients", the authors proposed and studied a new kinetic proofreading (KP) model/scheme based on having a spatial gradient of the substrate (both "correct" and "wrong" ones) and the diffusive transport of the substrate-bound enzyme molecules to a spatially localized production site. The authors did an excellent job in explaining their new model and its connection and difference with regards to the classical Hopfield-Ninos KP mechanism. The key insight is that with spatial inhomogeneity, e.g., in the presence of a persistent spatial gradient for the enzyme or the substrate, one can consider spatial location as a state-variable. By having the substrate and product (or production site) at different spatial locations, these spatial degrees of freedom of the enzyme, i.e., enzymes at different physical location, can be considered as the intermediate states that are necessary for kinetic proofreading – each intermediate state contributes a certain probability for error-correction. In the original Hopfield-Ninos KP scheme, the intermediate state is provided by additional enzyme(s), whereas in this new KP scheme, it depends on having a spatial gradient, which the authors argue is more tunable. The reviewers were enthusiastic about the theoretical model presented in this study because of its simplicity and elegance. However, the reviewers have also raised serious concerns (see Essential Revisions for detail) that need to be addressed in order to consider the manuscript further for publication in Life. In summary, the panel feels that discussion of possible biological example(s) where this novel type of proofreading may be occurring would significantly improve the manuscript's appeal to a broad audience. In addition, the reviewers ask for more explicitly explanation of the effect of enzymatic catalysis rates, and discussion of the full dissipation cost in the revised manuscript.

Essential revisions:

1) The major concern of the reviewer panel is how relevant this mechanism is for realistic biological systems. The original Hopfield-Ninos KP mechanism was motivated by specific and important biological problems (puzzles), namely the unusually high fidelity in biochemical synthesis process (in comparison with its equilibrium value). In this manuscript, the theory is developed without specific biological system or specific biological question in mind. It is true that spatial gradient exists across biological systems and the authors also showed that typical kinetic rates may fall in the functional range of this new gradient-dependent kinetic proofreading mechanism. But, what is the function of the original system that such a kinetic proofreading process can help improve? Is it biochemical synthesis? Do the authors envision "correct" and "wrong" biomolecules being produced at the production site (x=L) like in the original setting of Hopfield-Ninos? Or is it signaling like in the T-cell signaling case? If so, do the authors envision that both the correct signaling molecule and the incorrect signaling molecule have a spatial gradient and they can both be carried by the same enzyme to their functional sites? The panel is not asking a detailed comparison with a specific system, but a known biological phenomenon that may be explained by this new mechanism would help motivate the mostly biologist audience of *eLife*. Furthermore, a connection to a specific biological system could also lead to testable predictions that would ultimately verify (or falsify) the existence of this mechanism.

2) The entire manuscript assumes that catalysis is negligible and thus need not be explicitly modeled in solving for the steady-state distributions. How would incorporating a boundary condition at the right that involves non-negligible catalysis change (even qualitatively) your findings? To be more specific, there is a production r for the enzymatic reaction at x=L where the enzyme is active. However, the effect of this reaction, which change ESE+P, is not considered in the model equations (Equations 1-3). Is it because r is considered to be small? If so, smaller than what? Since speed is directly related to r, how does the value of r affect the speed and the speed-accuracy trade-off?

3) When quantifying the energetic costs, the main text solely focuses on the cost of counteracting the enzyme binding substrate, diffusing, and releasing. The appendix explores some theory for the other cost of maintaining the substrate gradients, but without reporting any absolute numbers. For the biologically plausible kinase/phosphatase substrate-maintenance mechanism explored in the main text, how does its cost compare to the cost that you study quantitatively in the main text? Specifically, where does Equation 8 come from? What's the physical meaning of *P*? The standard way to compute energy dissipation is by computing the entropy production rate S', which is well defined. Then by assuming the internal energy does not change with time in steady state, we equate energy dissipation with kT*S'. The form of entropy production rate is known and can be found in text book (such as those from T. Hill) and papers (e.g., those from H. Qian and collaborators; and from U. Seifert and collaborators), and the formula given in Equation 8 does not seem to be consistent with the known form of entropy production. In particular, for a given reaction with forward flux J+ and backward flux J-, the entropy production rate is: (J+-J-)ln(J+/J-), which can be easily shown to be positive definite and only = 0 when detailed balance J+=J- is satisfied.

4) The same concentration profiles are assumed for the right substrate R and the wrong substrate W. This is a strong assumption, could the authors consider the case where the concentration gradient length of the wrong substrate profile is larger than this length for the right substrate but still smaller that the distance L? They may calculate a series of the fidelity curves with increasing *λ_W_* and the same *λ_R_*. How will proofreading change?

---

## [Author Response]

Summary:In the manuscript by Galstyn et al. on "Proofreading through spatial gradients", the authors proposed and studied a new kinetic proofreading (KP) model/scheme based on having a spatial gradient of the substrate (both "correct" and "wrong" ones) and the diffusive transport of the substrate-bound enzyme molecules to a spatially localized production site. The authors did an excellent job in explaining their new model and its connection and difference with regards to the classical Hopfield-Ninos KP mechanism. The key insight is that with spatial inhomogeneity, e.g., in the presence of a persistent spatial gradient for the enzyme or the substrate, one can consider spatial location as a state-variable. By having the substrate and product (or production site) at different spatial locations, these spatial degrees of freedom of the enzyme, i.e., enzymes at different physical location, can be considered as the intermediate states that are necessary for kinetic proofreading – each intermediate state contributes a certain probability for error-correction. In the original Hopfield-Ninos KP scheme, the intermediate state is provided by additional enzyme(s), whereas in this new KP scheme, it depends on having a spatial gradient, which the authors argue is more tunable. The reviewers were enthusiastic about the theoretical model presented in this study because of its simplicity and elegance. However, the reviewers have also raised serious concerns (see Essential Revisions for detail) that need to be addressed in order to consider the manuscript further for publication in Life. In summary, the panel feels that discussion of possible biological example(s) where this novel type of proofreading may be occurring would significantly improve the manuscript's appeal to a broad audience. In addition, the reviewers ask for more explicitly explanation of the effect of enzymatic catalysis rates, and discussion of the full dissipation cost in the revised manuscript.

We are deeply grateful to the reviewers for the variety of very interesting suggestions and critiques that they have made. These remarks led the author team to several months of lively exchanges and precipitated a number of new and interesting calculations which are now in the paper or the supporting information. We have addressed all of the comments in detail, in many cases adding new calculations to the manuscript and we believe that the paper is much improved. We hope that the revised manuscript will now be viewed as suitable for publication.

The one comment that we wanted to address in a more circumspect fashion was the first comment concerning biological examples of our new proofreading hypothesis. We have several points to make here. First, in developing this new proofreading concept we were inspired by the ubiquitous phenomenon of allostery, the fact that many, many proteins change their state of activity upon binding to a relevant ligand. Further, many allosteric proteins are membrane bound. As a result, there are a plethora of examples where protein localization in conjunction with allostery provide a plausible basis for the kind of proofreading we envisage. As a result, although the reviewers wonder about biological examples which we describe below, in our view, the broad reach of the allostery phenomenon is to our minds an already strong plausibility argument for the kind of proofreading mechanisms we suggest here. A second more philosophical remark is simply to hope that the reviewers are open to the idea that there is something very powerful about theoretical ideas being ahead of experiments. For example, the idea of depletion forces was hypothesized by Asakura and Oosawa long before there was any data. With that in mind, we hope that the reviewers are open to the argument that this kind of interplay between theory and experiment in which a theoretical idea is ahead of the experiments is a potent tool for engendering new experiments. Indeed, the whole notion of positional information in the setting of morphogenesis is quite related to our work and we are inspired by the possibility of using synthetic biology approaches to explicitly construct the kind of mechanism we have hypothesized. In this era of synthetic biology, even if as yet there were no definitive natural examples of the mechanism we propose here, we are confident that this mechanism could be built using the tools of synthetic biology along the lines of the two papers that appeared in Science several weeks ago (Toda, et al. and Stapornwongkul, et al.) in which GFP was artificially used as a morphogen.

In summary, again, we are deeply grateful to the reviewers for many thoughtful and helpful comments. We have addressed all of them, though the question of biological examples is slightly nuanced.

Essential revisions:1) The major concern of the reviewer panel is how relevant this mechanism is for realistic biological systems. The original Hopfield-Ninos KP mechanism was motivated by specific and important biological problems (puzzles), namely the unusually high fidelity in biochemical synthesis process (in comparison with its equilibrium value). In this manuscript, the theory is developed without specific biological system or specific biological question in mind. It is true that spatial gradient exists across biological systems and the authors also showed that typical kinetic rates may fall in the functional range of this new gradient-dependent kinetic proofreading mechanism. But, what is the function of the original system that such a kinetic proofreading process can help improve? Is it biochemical synthesis? Do the authors envision "correct" and "wrong" biomolecules being produced at the production site (x=L) like in the original setting of Hopfield-Ninos? Or is it signaling like in the T-cell signaling case? If so, do the authors envision that both the correct signaling molecule and the incorrect signaling molecule have a spatial gradient and they can both be carried by the same enzyme to their functional sites? The panel is not asking a detailed comparison with a specific system, but a known biological phenomenon that may be explained by this new mechanism would help motivate the mostly biologist audience of eLife. Furthermore, a connection to a specific biological system could also lead to testable predictions that would ultimately verify (or falsify) the existence of this mechanism.

We thank the panel for urging us to propose more concrete biological examples where spatial proofreading could potentially be in play and for the questions about the implementation of our scheme. We have added several such examples in the Discussion section. As for the specific questions, in the processes discussed, it is indeed the case that the same enzyme/mediator protein transports both right and wrong substrates which either have a spatial gradient or are ideally localized at a membrane–bound compartment. The “product” of the reaction is the delivery of the substrate at the target site. This can either be the ultimate purpose of the pathway or be followed by biological synthesis. Specifically, the first example we discuss is related to spatially localized protein synthesis often seen in polarized, asymmetric cells. Designated ribonucleoproteins bind specific mRNAs near the cell nucleus and transport them to the localization site (e.g., the bud tip of a dividing cell, the lamellipodia or axonal growth cones) where synthesis occurs. mRNAs that are released during transport are subjected to degradation which prevents protein synthesis in the cytosol that, if it happened, could be toxic or deleterious to the cell (Parton, et al., 2014, Martin and Ephrussi, 2009). Another example is the non-vesicular transport of phospholipids between different membrane–bound compartments of the cell. This is achieved through lipid–transfer proteins that cycle between the donor and acceptor compartments and transfer specific lipids (Lev, 2010). Transport efficiency was mentioned in the review paper by Lev as an important performance metric dictated by the diffusion distance and we think it would be interesting to address the question of optimal architecture from the perspective of fidelity–transport efficiency (or, speed) trade-off. At the end of the Discussion section, we also mentioned a few other processes involving compartmentalized parts of the cell where our proposed scheme may be applicable. Experimental studies of these processes in in vivo and in vitro reconstituted settings in light of the signature features of the spatial proofreading mechanism will reveal if and to what extent it is used in cells. Lastly, we are very enthusiastic about the use of tools from synthetic biology to explicitly design and construct an in vivo example of our concept. Recent work on synthetic morphogen gradients (Toda, et al., 2020, Stapornwongkul, et al., 2020) foreshow these possibilities.

2) The entire manuscript assumes that catalysis is negligible and thus need not be explicitly modeled in solving for the steady-state distributions. How would incorporating a boundary condition at the right that involves non-negligible catalysis change (even qualitatively) your findings? To be more specific, there is a production r for the enzymatic reaction at x=L where the enzyme is active. However, the effect of this reaction, which change ESE+P, is not considered in the model equations (Equations 1-3). Is it because r is considered to be small? If so, smaller than what? Since speed is directly related to r, how does the value of r affect the speed and the speed-accuracy trade-off?

We thank the reviewer for raising this important point. We have now re-run our analysis with modified boundary conditions to account for finite catalysis rates (see Appendix 3). We showed that the performance of the spatial proofreading model depends on catalysis in the same qualitative way as with classical proofreading, namely, faster catalysis reduces the effectiveness of the final step of the proofreading cascade and the lowest error is achieved in the limit of slow catalysis. We derived exact analytical conditions for each regime and showed that fidelity reduction due to fast catalysis is bounded by a factor of *η*_eq_. We also showed that despite this reduction, higher catalysis rates in fact improve the Pareto–optimal front of the speed–fidelity trade-off. Specifically, to maximize speed for a given fidelity value, catalysis needs to be fast with a corresponding slowdown of diffusion. We comment on these new findings in the Results subsections “Slow Transport of Enzymatic Complex Enables Proofreading” and “Navigating the Speed–Fidelity Trade-Off”.

3) When quantifying the energetic costs, the main text solely focuses on the cost of counteracting the enzyme binding substrate, diffusing, and releasing. The appendix explores some theory for the other cost of maintaining the substrate gradients, but without reporting any absolute numbers. For the biologically plausible kinase/phosphatase substrate-maintenance mechanism explored in the main text, how does its cost compare to the cost that you study quantitatively in the main text? Specifically, where does Equation 8 come from? What's the physical meaning of P? The standard way to compute energy dissipation is by computing the entropy production rate S', which is well defined. Then by assuming the internal energy does not change with time in steady state, we equate energy dissipation with kT*S'. The form of entropy production rate is known and can be found in text book (such as those from T. Hill) and papers (e.g., those from H. Qian and collaborators; and from U. Seifert and collaborators), and the formula given in Equation 8 does not seem to be consistent with the known form of entropy production. In particular, for a given reaction with forward flux J+ and backward flux J-, the entropy production rate is: (J+-J-)ln(J+/J-), which can be easily shown to be positive definite and only = 0 when detailed balance J+=J- is satisfied.

In Appendix 6, subsection “Energy dissipation”, we calculated the total power dissipated in the kinase/phosphatase– based mechanics using a rough estimate for the dissipated energy per phosphorylation and dephosphorylation event (∼ 10 *k*_B_*T* each). We demonstrated that this cost exceeds our estimated lower bound on the proofreading cost (Figure 4 of the main text) as well as the minimum cost required for localizing substrates by roughly an order of magnitude for a wide region of the dissipation–fidelity trade-off curve (Appendix 6—figure 1), suggesting that for the purposes of spatial proofreading the energetic efficiency of the kinase/phosphatase–based mechanism is low. We mention this feature in the subsection “Proofreading by Biochemically Plausible Intracellular Gradients” of the main text.

In addition, we elaborated our discussion of the proofreading cost in Appendix 2, subsection “Derivation of the minimum dissipated power” and showed that our definition of power in Equation 8 of the main text in fact matches identically with the classical nonequilibrium thermodynamic definition expressed in terms of fluxes and thermodynamic forces (in the aforementioned subsection). The reason for their identity is the fact that driving forces are non-zero only for substrate binding/unbinding events and not for enzyme diffusion. Adding contributions from binding/unbinding events across the entire compartment leads to the proposed expression for power (Equation 8).

4) The same concentration profiles are assumed for the right substrate R and the wrong substrate W. This is a strong assumption, could the authors consider the case where the concentration gradient length of the wrong substrate profile is larger than this length for the right substrate but still smaller that the distance L? They may calculate a series of the fidelity curves with increasing λ_W_ and the same λ_R_. How will proofreading change?

In our main analysis, the assumption of equal concentration profiles for both substrates allows us to focus on discrimination due to the proofreading mechanism itself. We did not want to implicitly assume any discrimination of substrates other than the difference in their off-rates. By analogy, in classical proofreading models, one assumes that ATP hydrolysis (or any other energy consumption mechanism) itself does not discriminate between the substrates and that both substrates are present in equal amounts, even if neither is true in reality.

To understand how effects raised by the reviewers layer on top of the proofreading discrimination described in the main text, Appendix 4 now explores fidelity for unequal values of *λ_W_*and *λ_R_*. As anticipated, fidelity goes down with shallower gradients of wrong substrates. We note that, at least for the kinase/phosphatase-based gradient formation mechanism, it is in fact the *right* substrates that have a shallower gradient and not the wrong ones (e.g., see Appendix 6—figure 1C), which makes our assumption of equal concentration profiles a conservative one. This happens because wrong substrates unbind earlier in transport, and hence, are easier to localize than right substrates which are more likely to unbind closer to the production end. The curves plotted in Appendix 4—figure 1 demonstrate this in the *λ_W_ <λ_R_*region of the parameter space.